# Age, gender and socioeconomic patterns of awareness and usage of e-cigarettes across selected WHO region countries: evidence from the Global Adult Tobacco Survey

Sampurna Kundu [1,2] Subhojit Shaw [3] Junaid Khan [3]
Aparajita Chattopadhyay [4] Emerson Augusto Baptista [5] Balram Paswan[6]

For numbered affiliations see end of article.

**Correspondence to**
Aparajita Chattopadhyay;
aparajita@iipsindia.ac.in

## ABSTRACT

**Objectives** The study explores the awareness and e-cigarette use by demographic and socio-economic characteristics of selected 14 Global Adult Tobacco Survey (GATS) countries.
**Design** Cross-sectional.
**Setting** 14 countries.
**Participants** Surveyed population ≥15 years selected through multi-stage cluster sampling.
**Primary and secondary outcome measures** We selected 14 countries from 6 different WHO regions where GATS was conducted in different years during 2011–2017.
**Results** Awareness and usage of e-cigarette were highest in Greece and lowest in India. Females were less aware of e-cigarette across ages. The gender gap in awareness is wide in Greece post 50 years of age, while the gap is distinct in early ages in Kazakhstan and Qatar. The gender difference in use of e-cigarette was negligible in most of the countries except among the younger cohorts of Russia, Philippines Malaysia and Indonesia. Relatively higher prevalence of e-cigarette smoking among females in the older adult age was observed in some of the Asian countries like India. Multivariate analysis indicates that those who were younger, male, residing in urban areas, current tobacco smokers were more likely to use e-cigarette than their counterparts. Though prevalence of e-cigarette use increased with wealth and education, such pattern is not strong and consistent. Promotional advertisement plays important role in higher use of e-cigaratte. The predicted national prevalence of e-ciragette use was highest in Malaysia .
**Conclusions** E-cigarette use is more among urban adults, current smokers, males and in countries with promotional advertisement of e-cigarette. Area specific interventions are needed to understand the nature of e-cigarette use. Russia, Ukraine, Costa Rica and Mexico need better understanding to explore whether e-cigaratte use is an indulgence to new mode of addiction, as youth being highly likely to adopt this practice.

### STRENGTHS AND LIMITATIONS OF THIS STUDY

⇒ This study provides an important baseline of e-cigarette awareness and usage of selected GATS countries.
⇒ Among all the WHO study regions, Greece is the only country where use of e-cigarettes among females is markedly higher than males; while awareness and use of e-cigarette among Indian males and females are the lowest .
⇒ Prevalence of e-cigarette use is comparatively high in Russia and Malaysia, mainly among young adults and males.
⇒ A limitation to this study is that the estimates of e-cigarette awareness and its use are based on self-reported information.
⇒ As GATS data collection continues, the prevalence of e-cigarette use in each country can continue to be monitored and thus can be used to evaluate the existing policies over time.

widespread use may lead to future epidemic.[1–3] Subsequently, the popularity and use of e-cigarettes have increased over the years.[4] E-cigarettes and other battery-powered vaporisers were first launched in China, 2003 and later on entered the US market in 2007.[5] Surprisingly, during 2016, an estimated 54.6% of the e-cigarette users were also conventional cigarette smokers among US adults.[6] The use of e-cigarettes has evolved as a means of reducing the harmful effects of smoking while still delivering nicotine. E-cigarettes vaporise a liquid that contains nicotine for inhalation without the need of burning leaf materials. Despite its widespread adoption and high public exposure, there remains a disparity in the diffusion of e-cigarette awareness among different socio-demographic groups.[4]

There is no consensus in the previous literatures on the long-term effect or benefits of

## INTRODUCTION

The use of e-cigarettes has become common among youths and there are concerns that

**BMJ**

quitting smoking with e-cigarettes.[7] A few studies suggest that e-cigarettes can serve as a gateway for subsequent tobacco smoking.[8–11] According to a cross-sectional study in the USA, e-cigarette use is low among former smokers than the current adult smokers.[12] A study conducted by the US Food and Drug Administration revealed that e-cigarettes contain carcinogens, nitrosamines, diethylene glycol and other chemicals.[13] These chemicals have adverse effects on children, adolescents and pregnant mothers, contributing to cardiovascular diseases.[14–16] E-cigarettes are also available in varied flavours, making them more attractive and appealing to the youths.[17] For older adult smokers, e-cigarettes possess a beneficial health transition; however, amateur young adults who use e-cigarettes have a potential health risk. While nicotine itself is not a carcinogen, malignant diseases and neurodegeneration are suspected to result from nicotine.[14] A legitimate concern among the children is that if children (non-smokers) develops a nicotine addiction, they may start smoking cigarettes.[18] A recent study has suggested that young adults who use e-cigarettes had higher odds of conventional smoking initiation.[19] Furthermore, a report published by the National Academies of Sciences, Engineering, and Medicine[15] had found evidence of an increase in the risk of tobacco smoking due to e-cigarette use among young adults.[15] Hence, these trends pose major public health challenge that requires strict regulation pertaining to access of e-cigarettes.

Study conducted in the USA had shown that e-cigarette usage rates increased from 1.5% to 20.8% among youths during 2011–2018.[20] The use of e-cigarettes among youth adolescents has raised concerns about a new generation's lifelong addiction. An increase in the use of e-cigarettes among the youth, especially those in the higher economic sections is observed.[21 22] The diffusion and innovation theory by Everett Rogers, proposed in 1962, suggested that innovations are first appreciated by the upper class, followed by others.[23 24] E-cigarette use mimics low nicotine, reduces tar exposure, and is more aesthetically appealing than other forms of smoking, making it more attractive alternative with a higher rate of adaptation.[25] Therefore, an understanding of e-cigarette usage and its proximate determinants across different nations is critical. A handful of studies found that the likelihood of e-cigarette use was higher in the event of exposure to advertisements/promotions.[26–28]

Since 2014, the Institute for Global Tobacco Control, under Johns Hopkins Bloomberg School of Public Health, has been working globally across 120 nations to measure tobacco advertisement, promotion and evaluate the e-cigarette policy. In 2015, the WHO and World Bank Group called on WHO member countries to increase tobacco and related product taxes to prevent youths from initiating tobacco use.[29] In subsequent years, through numerous policies and programmes, the WHO offered to help fight the tobacco epidemic.[30] In collaboration, WHO member countries have developed mechanisms for tobacco cessation support. At the same time, the amount

and quality of scientific evidence have not been sufficient to determine whether e-cigarettes may help most smokers to quit or prevent tobacco smoking.[31] According to the WHO, member countries should regulate e-cigarettes to avoid promotion and intake by non-smokers, pregnant women and youths.[32]

A systematic review of literatures by Hartwell *et al*[4] revealed that the usage of e-cigarettes is higher among young adults, in higher socio-economic classes, and among individuals with a higher level of education.[4] The use of e-cigarettes is on the rise and a significant amount of research has been undertaken from a small selection of high-income countries.[1 2 8 12 33 34] In general, the use of e-cigarette is increasing among young adults as well as among the general adult population across many countries. Global Adult Tobacco Survey (GATS) in its member countries lately introduced a set of questions to collect the self-reported information on e-cigarette use and its awareness. Given the list of GATS countries, there exists no holistic research exploring the socio-economic and demographic determinants of e-cigarette awareness and its use. In this context, the present cross-country study explores the factors affecting adolescent and adult (population over 15 years) behaviour of e-cigarettes usage, employing the most recent round of the GATS datasets. From a socio-political perspective, this study is a timely contribution to identify the sub-population at higher risk of using e-cigarettes.

## DATA

GATS, an integral part of the Global Tobacco Surveillance System, is a nationally representative household survey started in 2008. The survey collects specific information on tobacco use and tracks key tobacco control indicators among non-institutionalised adults, 15 years of age or older in GATS countries. GATS uses country-specific stratified multi-stage cluster sampling design in which probability proportional to size, random selection methods are used to successively choose clusters in one or more steps to secure ample coverage of the target population. The sample of households is chosen in two or more stages, with sampling units in the first or second stage being well-defined geopolitical areas within the country. These areas are then randomly selected from a complete list of enumeration areas having no more than 250 households. At the final stage, households were surveyed from the randomly selected areas. From the selected household, one individual aged 15 years or older was randomly chosen to participate in the survey. The collection of information was carried out using electronic handheld devices. The overall response rate ranged from 64.4% to 98.5% across the selected counties in the study. For the first time since 2011, GATS introduced the questions on e-cigarette awareness and the current use of e-cigarettes.[35] Table 1 shows the GATS datasets being included in the study.

**Table 1** Description of the Global Adult Tobacco Survey dataset included in the study

| WHO regions | Country | Survey year | Households surveyed | Individuals interviewed | Response rate (%) |
|---|---|---|---|---|---|
| African region | Ethiopia | 2016 | 10875 | 10150 | 93.4 |
| | Senegal | 2017 | 4514 | 4347 | 97.0 |
| Eastern Mediterranean | Qatar | 2013 | 8571 | 8398 | 98.5 |
| European region | Greece | 2013 | 6600 | 4359 | 69.6 |
| | Kazakhstan | 2014 | 4611 | 4425 | 96.7 |
| | Russian Federation | 2016 | 11764 | 11458 | 98.2 |
| | Ukraine | 2017 | 14800 | 8298 | 64.4 |
| Region of the Americas | Costa Rica | 2015 | 9680 | 8607 | 89.2 |
| | Mexico | 2015 | 17765 | 14664 | 82.7 |
| South-East Asia region | Indonesia | 2011 | 8994 | 8305 | 94.3 |
| | India | 2017 | 84047 | 74037 | 92.9 |
| Western Pacific region | Malaysia | 2011 | 5112 | 4250 | 85.3 |
| | Philippines | 2015 | 13963 | 11644 | 88.4 |
| | Vietnam | 2015 | 9514 | 8996 | 95.8 |

Compiled by authors.

The surveys across the countries tracked down the awareness of e-cigarettes at different survey time points. In the survey questionnaire, e-cigarettes have been described as 'Electronic cigarettes include any product that uses batteries or other methods to produce a vapor which contains nicotine.' The description also includes various other names such as e-cigarette, vape-pen, e-shisha, e-pipes. The respondents were then asked, 'Have you ever heard of e-cigarettes?'. Those who responded 'yes' were considered to be aware of e-cigarettes. Those individuals who were aware of e-cigarettes were further asked to respond if he/she had currently used e-cigarettes on a daily basis, less than daily or not at all. Those who responded as 'daily' or 'less than daily' were considered current users of e-cigarettes. In this study, we selected those 14 countries that collected the specific information on e-cigarette use and its awareness in their most recent rounds of the GATS (table 1).[36–38]

### Outcome variable

The key outcome variable of this study was the respondent's current e-cigarette use status and was defined as a dichotomous (yes/no) variable. The variable was coded as '1' if the respondent answered as currently using e-cigarettes on 'daily' or 'less than daily' basis and '0' otherwise.

### Independent variables

Among the independent variables, the study used various socio-economic and demographic characteristics of the study population such as gender, age, residence, wealth index, education and occupation. According to the sample distribution, age was recoded into four age groups: 15–24, 25–44, 45–64 and above 64. Education was categorised as: no formal education/less than primary, completed primary/less than secondary, completed secondary or completed high school, and completed college or university or above. The 'occupation' variable included two categories, 'employed' and 'non-employed/unemployed'.

Wealth quintile variable has been computed in this study to measure the economic status of the respondents, which is commonly used in the cross-sectional surveys.[39] Authors have followed the standard DHS framework to compute the wealth variable using principal component analysis.[40] This is a common computation method of creating wealth index variable where households are given scores based on the number and kinds of consumer goods it owned, ranging from electricity connectivity, flush toilet, fixed telephone, cell telephone, television, radio, car, refrigerator, scooter or motor cycle, washing machine, computer or laptop, internet connection, air conditioner and electric fan. Then each individual from the same household are ranked by the scores and the distribution was equally divided into five quintile categories. Of the bottom 20% of the population is identified as poorest, next 20% as poorer, and likewise the top 20% is identified as the richest. In the present study, we have further recoded the wealth quintile variable into three categories, such as poor, middle and rich.

The current tobacco use status was assessed by the question 'Do you currently smoke tobacco on a daily basis, less than daily, or not at all?'. People who responded 'daily' or 'less than daily' were considered as current tobacco smokers (coded 1) and 'not at all' were considered non-smokers (coded 0). The tobacco products used included manufactured cigarettes, cigars, pipes, hand-rolled cigarettes, kreteks, and water pipes.

The study used two types of noticing to advertisements-in stores and on the internet. The store advertisement

noticing was measured by the question 'In the past 30 days, have you noticed any advertisements or signs promoting cigarettes in stores where cigarettes are sold?' and, similarly, internet advertisement noticing was measured by the question 'In the past 30 days, have you noticed any advertisements or signs promoting cigarettes on the internet?'. Respondents who said 'Yes' were the individuals to notice the advertisements (coded 1), and coded 'No' as '0' otherwise.

### Statistical analysis

For each selected country, we calculated the prevalence of current use of e-cigarettes, as well as the percentage of awareness about e-cigarettes. Further, we used bivariate analysis in the form of $\chi^2$ test of independence and cross-tabulations as well. To examine the socio-economic and demographic determinants of e-cigarette use in the population of the respective countries, we employed country specific multivariate logistic regression. The general equation of the logistic equation is as follows:

$$\text{logit} \left( \frac{p_i}{1-p_i} \right) = \alpha + \beta_1 \left( \text{gender}_i \right) + \beta_2 \left( \text{age}_i \right) + \beta_3 \left( \text{residence}_i \right) + \beta_4 \left( \text{education}_i \right) + \beta_5 \left( \text{occupation}_i \right) + \beta_6 \left( \text{wealth index}_i \right) + \beta_7 \left( \text{CST}_i \right) + \beta_8 \left( \text{Ad}_{\text{stores}_i} \right) + \beta_9 \left( \text{Ad}_{\text{internet}_i} \right) + \varepsilon_i$$

Here, $\alpha$ is the intercept, $\beta_j$s are the coefficients and $\varepsilon_i$ is the error component. CST refers to current smoking of tobacco, 'Ad$_{\text{stores}}$' refers to the variable of noticing advertisements in stores and similarly 'Ad$_{\text{internet}}$' refers to noticing advertisements on internet.

Once the logistic model was fitted for a particular country, we estimated the predicted prevalence from the model fit which gave the average prevalence of e-cigarette use after adjusting for all other socio-economic and demographic factors for that country. The 'logit' package in STATA provides different features for postestimation. There are different postestimation commands available from the same package and we used the 'predict' command which by default estimates the probability of the positive outcome (Y=1). This 'predict' command generates a new variable to store the estimated probability for each of the study individual/entity. Thus, for each of the country specific fitted logistic model, we estimated the average probability of the positive outcome, which is actually the average probability of using e-cigarette in the study population and hence the average predicted prevalence of e-cigarette use in a particular country. Sample weights ('gatsweight' in the survey dataset) have been used throughout the analysis. STATA V.14.1 was used to analyse the datasets.

### Patient and public involvement

No patients involved.

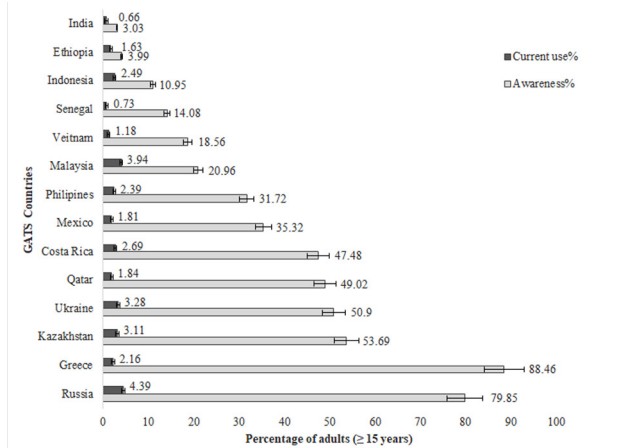

**Figure 1** The prevalence of e-cigarettes usage and awareness across selected GATS countries. Compiled by authors. Source: https://nccd.cdc.gov/GTSSDataSurveyResources/Ancillary/DataReports.aspx?CAID=2. GATS, Global Adult Tobacco Survey.

## RESULTS

### Awareness and current usage of e-cigarettes

Figure 1 shows awareness and current usage of e-cigarettes across selected countries. The percentage of population aware of e-cigarettes is highest in Greece (88.46%), followed by Russia (79.85%) and Kazakhstan (53.69%), that is, countries located in the European region. On the other extreme, India is the country with lowest awareness of e-cigarettes (3.03%), followed by Ethiopia (3.99%) and Indonesia (10.95%). Overall, the countries belonging to the European region shows a higher level of awareness in comparison to countries of other WHO regions.

Table 2 shows that awareness of e-cigarettes is higher among males and younger people. In addition, e-cigarette awareness is higher among those with more education (completed high school and higher), higher wealth index scores, noticing advertisement, living in urban areas, employed, and those who are currently smoking tobacco. In summary, the $\chi^2$ test (table 2) shows that there is a significant association between awareness of e-cigarettes and gender, age, place of residence, educational level, occupation, wealth quintile, current use of tobacco smoking and noticing of advertisement for almost all GATS countries studied. Exceptions are observed for Qatar, in Eastern Mediterranean region, where there is no significant association of awareness with noticing advertising in stores.

Regarding the prevalence of current use of e-cigarettes, it is generally low in majority of the selected GATS countries. The highest percentage of current usage of e-cigarettes is observed in Russia (4.39%), followed by Costa Rica (2.69%). On the other hand, countries like India (0.66%) and Senegal (0.73%) shows lowest current use (table 3). For Senegal (African region) and Vietnam (West Pacific region), although the prevalence is higher among men, there is no significant association observed between gender and current use of e-cigarettes. Few

**Table 2** Country specific bivariate estimation on awareness of e-cigarette use, Global Adult Tobacco Survey, 2011–2017

| | African region | | | | Eastern Mediterranean | | European region | | | | | | | | Regions of the Americas | | | | South-East Asia region | | | | Western Pacific region | | | | | |
| | Ethiopia | | Senegal | | Qatar | | Greece | | Kazakhstan | | Russian Federation | | Ukraine | | Costa Rica | | Mexico | | Indonesia | | India | | Malaysia | | Philippines | | Vietnam | |
| Predictors | % | χ² | % | χ² | % | χ² | % | χ² | % | χ² | % | χ² | % | χ² | % | χ² | % | χ² | % | χ² | % | χ² | % | χ² | % | χ² | % | χ² |
|---|---|---|---|---|---|---|---|---|---|---|---|---|---|---|---|---|---|---|---|---|---|---|---|---|---|---|---|---|
| **Gender** | | | | | | | | | | | | | | | | | | | | | | | | | | | | |
| Male | 5.23 | 58.90*** | 18.70 | 74.21*** | 62.80 | 835.79*** | 93.43 | 129.39*** | 64.06 | 174.80*** | 86.28 | 315.84*** | 58.04 | 216.63*** | 51.98 | 58.53*** | 40.93 | 193.02*** | 16.83 | 307.30*** | 4.14 | 313.67*** | 29.00 | 181.44*** | 36.36 | 122.65*** | 25.53 | 352.65*** |
| Female | 2.74 | | 9.73 | | 33.81 | | 83.67 | | 44.36 | | 74.52 | | 44.96 | | 42.94 | | 30.15 | | 5.09 | | 1.87 | | 12.43 | | 27.10 | | 12.01 | |
| **Age** | | | | | | | | | | | | | | | | | | | | | | | | | | | | |
| 15–24 | 2.96 | 19.69*** | 12.56 | 33.56*** | 48.78 | 78.31*** | 93.73 | 803.79*** | 59.56 | 350.55*** | 91.45 | 2400.00*** | 63.09 | 921.62*** | 51.93 | 207.56*** | 43.91 | 379.98*** | 14.24 | 141.08*** | 3.75 | 165.95*** | 24.11 | 139.43*** | 35.75 | 250.85*** | 20.97 | 143.86*** |
| 25–44 | 5.24 | | 16.30 | | 50.17 | | 95.22 | | 62.24 | | 92.26 | | 61.72 | | 51.72 | | 37.74 | | 12.53 | | 3.54 | | 26.17 | | 35.30 | | 21.64 | |
| 45–64 | 3.64 | | 14.81 | | 49.52 | | 35.21 | | 47.18 | | 80.17 | | 51.20 | | 43.56 | | 30.51 | | 7.89 | | 2.12 | | 13.06 | | 26.61 | | 15.64 | |
| Above 64 | 3.96 | | 5.09 | | 25.83 | | 62.17 | | 19.54 | | 44.11 | | 24.30 | | 28.91 | | 15.21 | | 1.72 | | 0.91 | | 7.57 | | 12.54 | | 7.90 | |
| **Residence** | | | | | | | | | | | | | | | | | | | | | | | | | | | | |
| Urban | 8.27 | 176.61*** | 24.22 | 362.19*** | NA | NA | 90.43 | 73.74*** | 63.42 | 225.18*** | 81.98 | 211.83*** | 55.89 | 247.33*** | 51.80 | 248.05*** | 41.17 | 1100.00*** | 15.34 | 181.09*** | 5.12 | 663.54*** | 23.05 | 38.55*** | 43.83 | 576.93*** | 27.51 | 195.41*** |
| Rural | 2.62 | | 3.97 | | NA | | 83.25 | | 40.97 | | 73.46 | | 39.58 | | 35.21 | | 13.71 | | 6.53 | | 1.93 | | 15.55 | | 21.11 | | 14.00 | |
| **Education** | | | | | | | | | | | | | | | | | | | | | | | | | | | | |
| No formal education | 2.42 | 348.46*** | 2.97 | 782.24*** | 15.95 | 328.34*** | 49.85 | 809.60*** | 36.84 | 221.97*** | 33.44 | 1100.00*** | 49.25 | 503.06*** | 28.48 | 462.22*** | 9.40 | 1500.00*** | 1.40 | 575.35*** | 0.70 | 2300.00*** | 6.56 | 292.71*** | 13.16 | 840.25*** | 4.11 | 725.71*** |
| Completed primary | 3.26 | | 26.34 | | 39.68 | | 74.55 | | 38.70 | | 56.29 | | 15.07 | | 37.84 | | 21.22 | | 5.74 | | 2.13 | | 13.21 | | 21.25 | | 9.57 | |
| Completed secondary | 5.07 | | 35.98 | | 48.53 | | 94.43 | | 56.43 | | 79.08 | | 49.57 | | 50.04 | | 42.39 | | 16.08 | | 4.27 | | 23.27 | | 23.10 | | 20.76 | |
| Completed college/university | 13.99 | | 59.68 | | 56.52 | | 96.50 | | 64.02 | | 89.33 | | 61.28 | | 68.05 | | 59.20 | | 29.41 | | 10.07 | | 43.28 | | 50.30 | | 39.90 | |
| **Occupation** | | | | | | | | | | | | | | | | | | | | | | | | | | | | |
| Non-employed | 2.35 | 25.56*** | 12.07 | 25.57*** | 36.00 | 604.03*** | 82.72 | 291.99*** | 41.10 | 309.02*** | 65.19 | 1400.00*** | 41.29 | 538.77*** | 40.94 | 184.19*** | 30.38 | 247.09*** | 8.89 | 28.85*** | 2.66 | 47.21*** | 15.58 | 105.77*** | 28.08 | 38.85*** | 16.93 | 16.32*** |
| Employed | 5.59 | | 16.23 | | 59.14 | | 96.58 | | 65.11 | | 90.47 | | 61.69 | | 54.73 | | 40.02 | | 12.16 | | 3.37 | | 25.19 | | 34.06 | | 19.15 | |
| **Wealth index** | | | | | | | | | | | | | | | | | | | | | | | | | | | | |
| Rich | 7.76 | 269.44*** | 25.74 | 248.38*** | NA | NA | 93.57 | 398.44*** | 66.06 | 305.87*** | 86.09 | 614.97*** | 61.24 | 731.61*** | 54.79 | 271.09*** | 48.67 | 1500.00*** | 18.49 | 432.58*** | 5.86 | 1400.00*** | 27.46 | 185.19*** | 44.73 | 847.92*** | 31.08 | 615.73*** |
| Middle | 2.66 | | 9.47 | | NA | | NA | | 50.71 | | 75.80 | | 46.04 | | 42.99 | | 25.96 | | 8.07 | | 1.79 | | 19.35 | | 25.94 | | 16.54 | |
| Poor | 2.85 | | 2.60 | | NA | | 74.90 | | 37.72 | | 69.26 | | 32.09 | | 37.77 | | 14.80 | | 3.15 | | 1.28 | | 9.02 | | 15.97 | | 8.11 | |
| **Currently smoke tobacco** | | | | | | | | | | | | | | | | | | | | | | | | | | | | |
| No | 3.75 | 70.39*** | 13.66 | 7.17*** | 44.41 | 529.69*** | 83.93 | 219.58*** | 48.35 | 184.62*** | 74.97 | 493.53*** | 46.72 | 263.43*** | 45.00 | 194.59*** | 31.46 | 592.80*** | 7.75 | 163.63*** | 2.88 | 33.89*** | 17.60 | 101.71*** | 29.43 | 63.30*** | 16.91 | 67.99*** |
| Yes | 10.10 | | 21.49 | | 82.68 | | 96.79 | | 72.16 | | 91.07 | | 65.02 | | 72.79 | | 55.03 | | 16.94 | | 4.24 | | 32.11 | | 39.52 | | 24.25 | |
| **Advertisements in stores** | | | | | | | | | | | | | | | | | | | | | | | | | | | | |
| No | 3.95 | 19.15*** | 12.44 | 64.13*** | 48.91 | 0.59 | 85.38 | 120.96*** | 53.75 | 0.01*** | 79.33 | 37.97*** | 50.01 | 39.73*** | 46.47 | 23.91*** | 33.08 | 109.49*** | 7.93 | 124.73*** | 2.60 | 327.46*** | 19.55 | 22.53*** | 28.85 | 66.40*** | 17.15 | 99.34*** |
| Yes | 15.38 | | 31.35 | | 49.93 | | 96.26 | | 53.27 | | 89.22 | | 56.49 | | 57.00 | | 39.22 | | 14.54 | | 7.17 | | 27.00 | | 35.94 | | 33.49 | |
| **Advertisements on internet** | | | | | | | | | | | | | | | | | | | | | | | | | | | | |
| No | 3.96 | 11.31*** | 13.62 | 86.62*** | 48.20 | 6.90*** | 88.13 | 16.81*** | 52.67 | 25.61*** | 78.67 | 130.31*** | 50.60 | 18.14*** | 46.45 | 61.59*** | 33.08 | 360.31*** | 10.45 | 124.78*** | 2.93 | 245.11*** | 20.28 | 24.79*** | 29.60 | 312.69*** | 18.50 | 11.14*** |
| Yes | 12.85 | | 57.98 | | 54.59 | | 99.02 | | 67.23 | | 93.90 | | 57.84 | | 64.45 | | 52.11 | | 36.71 | | 11.39 | | 35.82 | | 64.92 | | 25.96 | |

*p<0.001; **p<0.01; ***p<0.05.
NA denotes data not available or survey not available.

**Table 3** Country specific bivariate estimation on current use of e-cigarettes, Global Adult Tobacco Survey, 2011–2017

| | African region | | | | Eastern Mediterranean | | European region | | | | | | | | Regions of the Americas | | | | South-East Asia region | | | | Western Pacific region | | | | | |
| | Ethiopia | | Senegal | | Qatar | | Greece | | Kazakhstan | | Russian Federation | | Ukraine | | Costa Rica | | Mexico | | Indonesia | | India | | Malaysia | | Philippines | | Vietnam | |
| Predictors | % | $\chi^2$ | % | $\chi^2$ | % | $\chi^2$ | % | $\chi^2$ | % | $\chi^2$ | % | $\chi^2$ | % | $\chi^2$ | % | $\chi^2$ | % | $\chi^2$ | % | $\chi^2$ | % | $\chi^2$ | % | $\chi^2$ | % | $\chi^2$ | % | $\chi^2$ |
|---|---|---|---|---|---|---|---|---|---|---|---|---|---|---|---|---|---|---|---|---|---|---|---|---|---|---|---|---|
| Gender | | | | | | | | | | | | | | | | | | | | | | | | | | | | |
| Male | 1.79 | 1.1 | 0.89 | 0.04 | 2.51 | 12.51*** | 1.81 | 0.39 | 3.9 | 8.05** | 6.29 | 80.19*** | 4.31 | 7.26*** | 3.1 | 4.68** | 2.62 | 9.49*** | 3.24 | 6.94*** | 0.81 | 5.93** | 5.53 | 8.20** | 3.7 | 25.79*** | 1.38 | 1.49 |
| Female | 1.34 | | 0.46 | | 0.48 | | 2.54 | | 2.09 | | 2.55 | | 2.18 | | 2.18 | | 0.8 | | 0 | | 0.31 | | 0 | | 0.64 | | 0.77 | |
| Age | | | | | | | | | | | | | | | | | | | | | | | | | | | | |
| 15–24 | 2.25 | 0.86 | 1.89 | 3.29 | 0.93 | 4.07 | 0 | 12.39*** | 3.15 | 4.92 | 10.51 | 180.44*** | 7.6 | 63.71*** | 4.87 | 16.51*** | 3.57 | 24.53*** | 1.31 | 1.54 | 0.51 | 6.59* | 3.94 | 6.8 | 1.84 | 1.53 | 0.66 | 2.2 |
| 25–44 | 1.03 | | 0.34 | | 2.01 | | 2.99 | | 3.87 | | 5.2 | | 3.86 | | 2.16 | | 1.28 | | 2.47 | | 0.73 | | 2.27 | | 2.8 | | 1.26 | |
| 45–64 | 3.03 | | 0 | | 2.55 | | 2.79 | | 1.99 | | 1.91 | | 1.44 | | 1.89 | | 0.66 | | 4.56 | | 0.61 | | 0 | | 2.24 | | 1.75 | |
| Above 64 | 0 | | 0 | | 0 | | 1.25 | | 0.27 | | 0.54 | | 1.04 | | 0.45 | | 0.25 | | 0 | | 1.51 | | 0 | | 2.42 | | 0 | |
| Residence | | | | | | | | | | | | | | | | | | | | | | | | | | | | |
| Urban | 2.37 | 0.01 | 0.77 | 0.99 | NA | NA | 2.14 | 0.06 | 3.65 | 4.58** | 4.86 | 38.16*** | 3.36 | 3.41 | 2.95 | 6.89*** | 1.85 | 1.76 | 2.75 | 0.35 | 0.93 | 0.73 | 4.6 | 4.24** | 3.07 | 6.99*** | 0.57 | 4.04** |
| Rural | 0.89 | | 0.52 | | NA | | 2.22 | | 2.02 | | 2.8 | | 3.03 | | 1.58 | | 1.35 | | 1.87 | | 0.28 | | 1.4 | | 1.15 | | 1.79 | |
| Education | | | | | | | | | | | | | | | | | | | | | | | | | | | | |
| No formal education | 2.05 | 0.19 | 0.2 | 0.22 | 5.1 | 0.72 | 2.43 | 21.95*** | 2.13 | 2.85 | 0 | 4.37*** | 0 | 1.38 | 1.41 | 1.8 | 2.04 | 6.26 | 0 | 1.88 | 1.09 | 1.43 | 6.44 | 3.76 | 0 | 1.29 | 0 | 3.04 |
| Completed primary | 1.69 | | 1.12 | | 1.5 | | 0.53 | | 2.32 | | 3.96 | | 0.37 | | 2.07 | | 0.72 | | 4.16 | | 0.19 | | 4.36 | | 0 | | 1.63 | |
| Completed secondary | 1.55 | | 0 | | 1.64 | | 1.56 | | 2.81 | | 4.24 | | 3.07 | | 3.75 | | 2.09 | | 2.12 | | 0.82 | | 4.81 | | 2.3 | | 1.03 | |
| Completed college/university | 1.24 | | 0.72 | | 1.93 | | 4.47 | | 3.81 | | 4.69 | | 3.88 | | 1.32 | | 1.59 | | 2.85 | | 0.66 | | 1.59 | | 2.15 | | 1.42 | |
| Occupation | | | | | | | | | | | | | | | | | | | | | | | | | | | | |
| Non-employed | 0.71 | 1.89 | 0.37 | 0.04 | 0.79 | 7.59*** | 1.23 | 14.58*** | 2.16 | 3.87** | 3.55 | 17.07*** | 2.09 | 9.40*** | 2.99 | 1.85 | 2.14 | 0.43 | 1.6 | 0.9 | 0.21 | 7.09*** | 0.49 | 3.42 | 1.63 | 5.48** | 0 | 4.78** |
| Employed | 2.02 | | 1.02 | | 2.34 | | 3.29 | | 3.66 | | 4.82 | | 4.18 | | 2.44 | | 1.57 | | 2.87 | | 0.99 | | 5.62 | | 2.79 | | 1.55 | |
| Wealth index | | | | | | | | | | | | | | | | | | | | | | | | | | | | |
| Rich | 1.59 | 0.36 | 0.92 | 5.55** | NA | NA | 2.4 | 5.14** | 3.63 | 3.94 | 4.82 | 16.79*** | 3.6 | 15.14 | 2.91 | 0.33 | 1.98 | 3.63 | 2.27 | 1.05 | 0.6 | 2.52 | 3.3 | 0.25 | 2.91 | 4.81 | 1.02 | 0.09 |
| Middle | 0.03 | | 0 | | NA | | NA | | 1.81 | | 2.49 | | 3.96 | | 2.68 | | 1.98 | | 4.21 | | 0.42 | | 6.57 | | 1.55 | | 1.66 | |
| Poor | 2.03 | | 2.96 | | NA | | 1.36 | | 2.66 | | 4.18 | | 1.28 | | 2.15 | | 0.49 | | 1.05 | | 1.06 | | 3.07 | | 1.09 | | 1.26 | |
| Currently smoke tobacco | | | | | | | | | | | | | | | | | | | | | | | | | | | | |
| No | 1.48 | 11.28*** | 0.72 | 2.3 | 0.37 | 173.69*** | 1.31 | 11.05*** | 0.83 | 87.43*** | 1.77 | 219.38*** | 1.91 | 68.28*** | 1.69 | 117.02*** | 0.93 | 45.92*** | 0.43 | 12.08*** | 0.36 | 59.09*** | 0.38 | 30.27*** | 0.59 | 80.01*** | 0.14 | 23.75*** |
| Yes | 3.11 | | 0.9 | | 7.62 | | 3.36 | | 8.41 | | 9.33 | | 6.61 | | 8.97 | | 4.38 | | 4.25 | | 2.37 | | 10.42 | | 6.97 | | 3.67 | |
| Advertisements in stores | | | | | | | | | | | | | | | | | | | | | | | | | | | | |
| No | 1.66 | 0.16 | 0.2 | 0.15 | 1.71 | 1.33 | 1.68 | 12.64*** | 3.02 | 1.03 | 4.21 | 9.37*** | 2.91 | 14.41*** | 2.73 | 0.77 | 1.64 | 1.27 | 2.32 | 1.05 | 0.6 | 0.6 | 3.71 | 0.07 | 2.29 | 0.87 | 1.01 | 0.48 |
| Yes | 0 | | 2.96 | | 2.92 | | 3.23 | | 3.68 | | 7.17 | | 5.35 | | 2.41 | | 2.12 | | 2.59 | | 0.87 | | 4.64 | | 2.51 | | 2.09 | |
| Advertisements on internet | | | | | | | | | | | | | | | | | | | | | | | | | | | | |
| No | 1.65 | 0.13 | 0.77 | 0.15 | 1.83 | 0.03 | 2.11 | 2.33 | 3.17 | 0.15 | 3.99 | 24.26*** | 3.27 | 4.53** | 2.67 | 0 | 1.68 | 4.99** | 2.65 | 1.49 | 0.66 | 0.04 | 3.85 | 0.75 | 2.05 | 11.94*** | 1.17 | 3.26** |
| Yes | 0 | | 0 | | 1.9 | | 3.65 | | 2.55 | | 8.31 | | 3.52 | | 2.89 | | 2.43 | | 0 | | 0.75 | | 5.11 | | 4.83 | | 1.57 | |

***p<0.001, **p<0.01, *p<0.05.
NA denotes data not available or cases not available.

countries in European and American regions reflect high usage among the younger age groups (15–24 age): Russia (10.5%), Ukraine (7.6%) and Costa Rica (4.87%). In the American region, Costa Rica has the higher usage overall, but it is significantly associated with gender, age, residence and current tobacco smoking only. In addition, few countries show high usage in older age groups (64 age and above): Philippines (2.42%), India (1.51%) and Greece (1.25%), in the Western Pacific, South-East Asia and European region, respectively (table 3). In terms of age groups, there are significant differences observed in current e-cigarette use across the countries of Costa Rica, Greece, Mexico and Ukraine. Except Greece and Russia (European region), none of the countries had any significant association of current use of e-cigarettes with educational level, although people with higher levels of education make more use of e-cigarettes (completed college or university, Greece: 4.5% and Russia: 4.7%). In all countries, the prevalence of current e-cigarette use is observed to be higher among those who smoke tobacco than the non-smokers. It is also found that current usage of smoking tobacco has a significant association with current use of e-cigarettes in the population across all the countries, except for Senegal (users: 0.9%, non-users: 0.72%) (table 3).

The gender difference in awareness and use of e-cigarette curve of 14 countries are given in figures 2 and 3. In general, awareness decreases with age, except Russia where the percent adult who are aware of e-cigarette increases sharply after age 50. Females are less aware of e-cigarette across ages. Sharpest decline of awareness of e-cigarette is visible in Greece (mainly for females), Kazakhstan and Qatar (mainly for males). The gender gap in awareness in wide in Greece, post 50 years, while the gap is distinct in early ages in countries like Kazakhstan and Qatar. Male tends to use e-cigarette more than females across ages (figure 2). The gender gap in use of e-cigarette is negligible in most of the countries except among the younger cohorts of Russia, Philippines, Malaysia and Indonesia where prevalence of use is distinctly more among the young males. Point to be highlighted here is the relatively higher prevalence of e-cigarette smoking among females in the older adult age (50+) observed in some of the Asian countries, i.e. India, Philippines, Qatar. However, smaller sample size of 50+ population in the data may be considered for a cautious interpretation (figure 3).

### Country specific multivariate logistic estimation

Table 4 shows the country-specific multivariate logistic estimation of current use of e-cigarettes. In the European

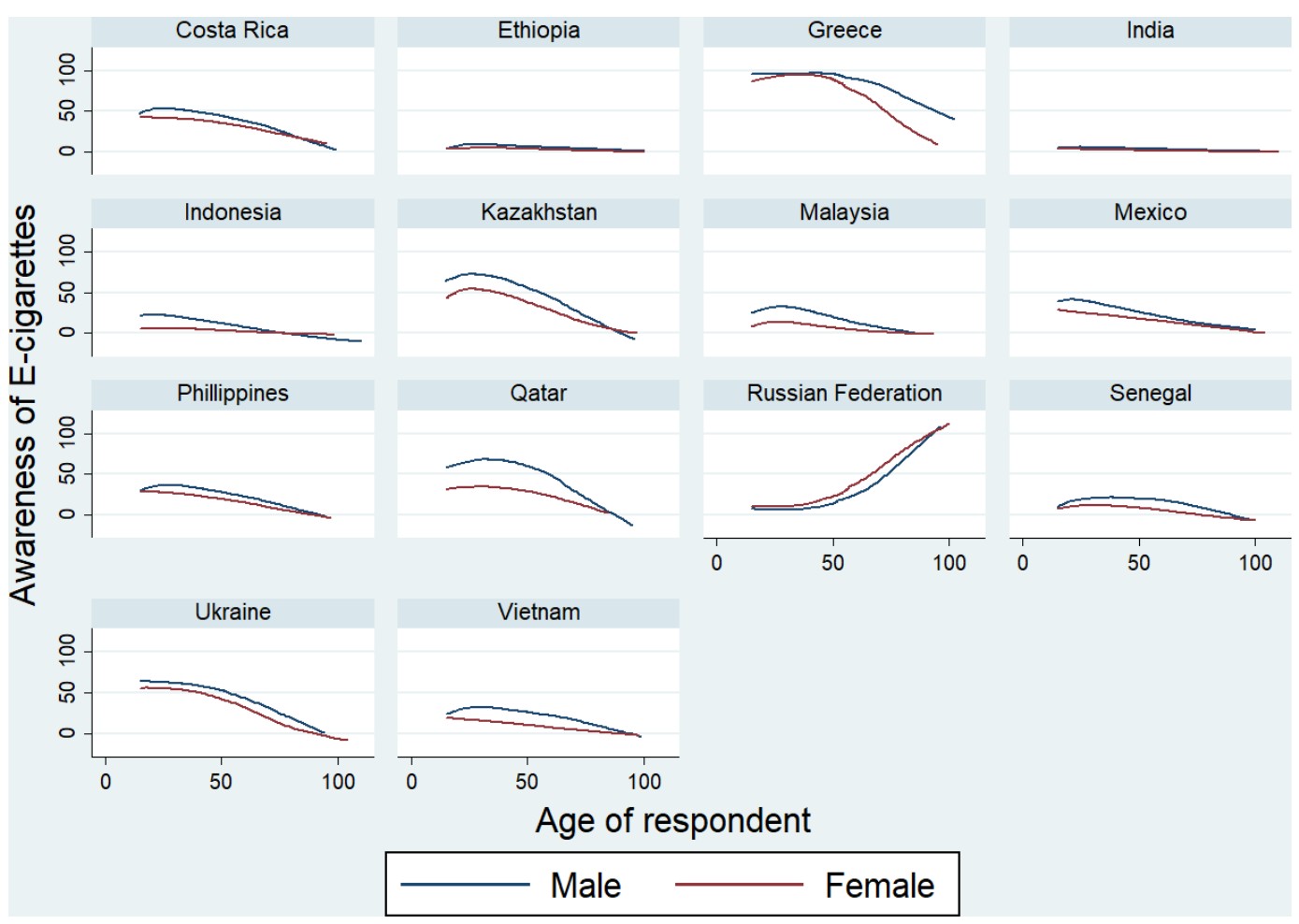

**Figure 2** Age-sex wise prevalence of awareness of e-cigarettes across selected WHO countries.

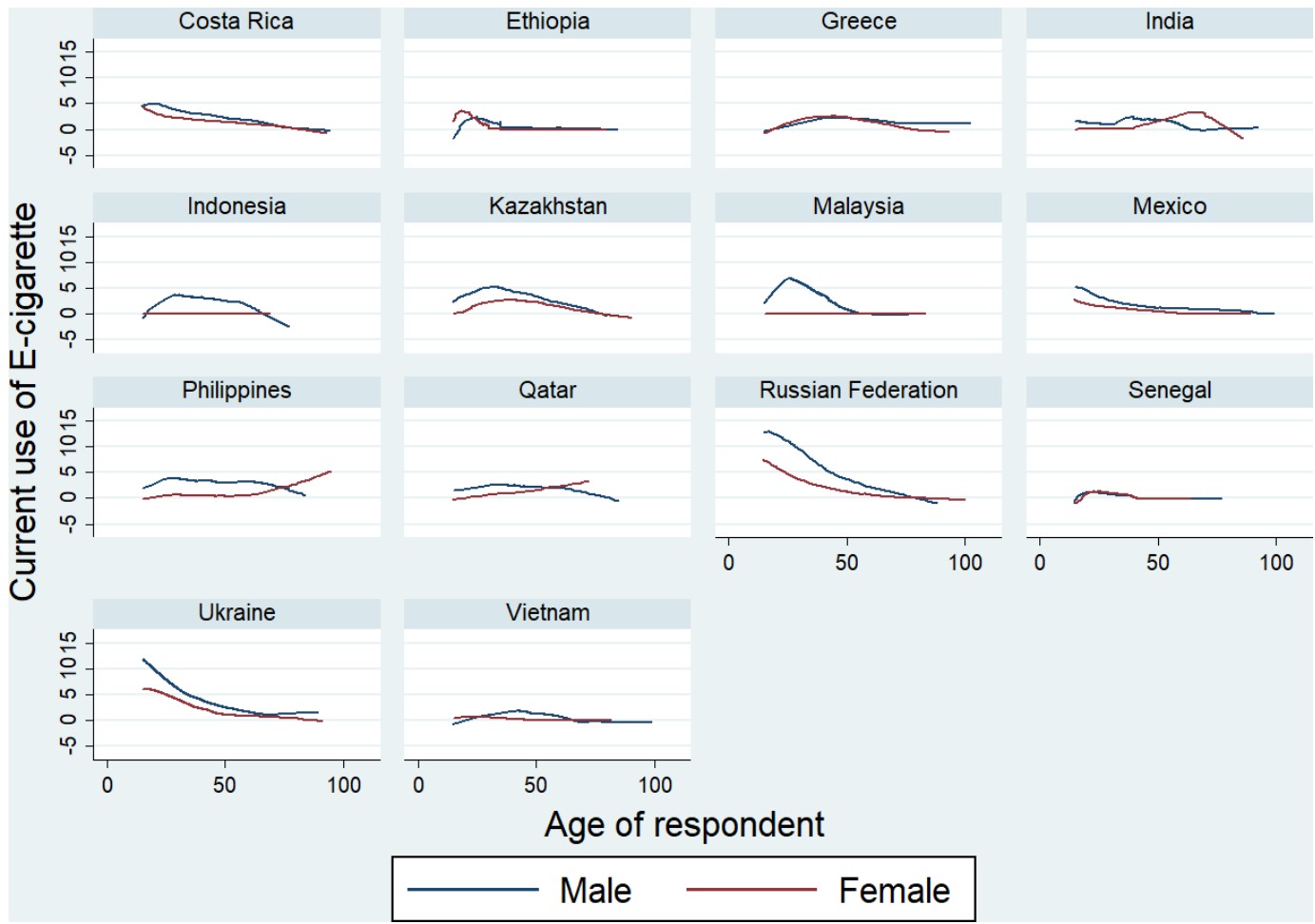

**Figure 3** Age-sex wise prevalence of current use of e-cigarettes across selected WHO countries.

region, Greece, where awareness of e-cigarettes is the highest, females are 2.27 times more likely to use e-cigarettes than males. Females in Qatar (Eastern Mediterranean region), Russia (European region), Mexico (American region) and Philippines (Western Pacific region), are 0.42, 0.63, 0.37 and 0.44 times less likely respectively to use e-cigarettes than males. The odds of e-cigarette use show a gradual decrease with increasing age in the countries like Russia, Costa Rica, Mexico and Ukraine. It is also observed that there is a significant effect of place of residence on use of e-cigarettes in few of the countries, such as Ethiopia (adjusted odds ratio (AOR): 0.01, 95% CI 0.00 to 0.15), Russia (AOR: 0.59, 95% CI 0.44 to 0.80), Malaysia (AOR: 0.25, 95% CI 0.06 to 0.95), Philippines (AOR: 0.40, 95% CI 0.21 to 0.78) where people in rural areas are less likely to use e-cigarettes than those residing in the urban areas. Whereas in Vietnam, people from rural areas are 4.36 times more likely to use e-cigarettes. In terms of occupation, the employed people of Ukraine are 1.64 times more likely to use e-cigarettes than the unemployed people. Countries like Russia and Greece, which are in the European region, have shown significant variations in the use of e-cigarettes by levels of education. Household's economic well-being (wealth index) has not shown any significant effect on e-cigarette

usage for majority of the study countries. In contrast, poor people of Mexico (AOR: 0.17, 95% CI 0.07 to 0.54) are less likely to use e-cigarettes than the rich. It is also found that the current tobacco smokers are more likely to use e-cigarettes than non-smokers across all the countries, except for Indonesia (South East Asia region), Senegal and Ethiopia (African region). Philippines (AOR: 2.96, 95% CI 1.41 to 6.21) demonstrates a significant association between noticing advertisement on internet and usage of e-cigarettes at 1% level of significance.

**Predicted average prevalence of e-cigarettes use in the countries**
From the country-specific logistic regression (table 4), we predicted the average national prevalence for the respective countries. When adjusted for the socio-economic and demographic factors we found that Malaysia carried the highest burden of e-cigarette use (39/1000 population) during 2011. Countries like the Russian Federation had a prevalence of 37/1000 population during 2016 whereas in Ethiopia, the prevalence was 16/1000 population in the same year. Qatar from the Eastern Mediterranean region and Greece from the European region showed almost an equal prevalence of e-cigarette use during 2013. On the other hand, Kazakhstan during 2014 and Ukraine during

**Table 4** Country specific multivariate logistic regression for current use of e-cigarette, Global Adult Tobacco Survey, 2011–2017

| | African region | | Eastern Mediterranean | European region | | | | Regions of the Americas | | South-East Asia region | | Western Pacific region | | |
| --- | --- | --- | --- | --- | --- | --- | --- | --- | --- | --- | --- | --- | --- | --- |
| Predictors | Ethiopia AOR (95% CI) | Senegal AOR (95% CI) | Qatar AOR (95% CI) | Greece AOR (95% CI) | Kazakhstan AOR (95% CI) | Russian Federation AOR (95% CI) | Ukraine AOR (95% CI) | Costa Rica AOR (95% CI) | Mexico AOR (95% CI) | Indonesia AOR (95% CI) | India AOR (95% CI) | Malaysia AOR (95% CI) | Philippines AOR (95% CI) | Vietnam AOR (95% CI) |
| **Gender** | | | | | | | | | | | | | | |
| Male# | 1 | 1 | 1 | 1 | 1 | 1 | 1 | 1 | 1 | 1 | 1 | 1 | 1 | 1 |
| Female | 0.38 (0.07 to 1.99) | 0.53 (0.04 to 6.33) | 0.42** (0.18 to 0.98) | 2.27* (1.12 to 4.6) | 1.7 (0.9 to 3.22) | 0.62** (0.46 to 0.85) | 0.83 (0.47 to 1.46) | 0.91 (0.48 to 1.72) | 0.37* (0.17 to 0.81) | NA | 0.84 (0.14 to 5.09) | NA | 0.43 (0.16 to 1.13) | [46.48]*** (5.77 to 374.2) |
| **Age** | | | | | | | | | | | | | | |
| 15–24# | 1 | 1 | 1 | 1 | 1 | 1 | 1 | 1 | 1 | 1 | 1 | 1 | 1 | 1 |
| 25–44 | 0.16 (0.02 to 1.12) | 0.03 (0 to 1.84) | 1.18 (0.46 to 3.05) | 0.83 (0.21 to 3.25) | 0.53 (0.24 to 1.17) | 0.31*** (0.22 to 0.44) | 0.31*** (0.17 to 0.57) | 0.42** (0.23 to 0.75) | 0.35** (0.17 to 0.72) | 2.23 (0.46 to 10.69) | 0.47 (0.07 to 3.18) | 0.51 (0.08 to 3.23) | 1.04 (0.5 to 2.14) | 1.28 (0.33 to 4.99) |
| 45–64 | 0.50 (0 to 78.42) | NA | 1.47 (0.51 to 4.24) | 1.16 (0.28 to 4.88) | 0.29* (0.11 to 0.74) | 0.12*** (0.08 to 0.19) | 0.14*** (0.06 to 0.3) | 0.34** (0.17 to 0.7) | 0.17* (0.04 to 0.67) | 4.54 (1 to 20.61) | 0.35 (0.04 to 2.87) | NA | 0.87 (0.34 to 2.23) | 2.27 (0.52 to 9.94) |
| Above 64 | NA | NA | NA | NA | 0.05** (0.01 to 0.42) | 0.05*** (0.02 to 0.13) | 0.17** (0.05 to 0.58) | 0.10** (0.02 to 0.5) | 0.04* (0 to 0.51) | NA | 2.36 (0.32 to 17.4) | NA | 1.4 (0.23 to 8.47) | NA |
| **Residence** | | | | | | | | | | | | | | |
| Urban# | 1 | 1 | 1 | 1 | 1 | 1 | 1 | 1 | 1 | 1 | 1 | 1 | 1 | 1 |
| Rural | 4.66 (0.34 to 64.51) | | NA | 1.35 (0.63 to 2.91) | 0.82 (0.44 to 1.55) | 0.59** (0.44 to 0.80) | 0.89 (0.53 to 1.51) | 0.55 (0.3 to 1) | 0.9 (0.36 to 2.23) | 0.55 (0.15 to 2.06) | 0.19 (0.03 to 1.21) | 0.25* (0.06 to 0.95) | 0.40** (0.21 to 0.78) | 4.37* (1.08 to 17.73) |
| **Education** | | | | | | | | | | | | | | |
| No formal education# | 1 | 1 | 1 | 1 | 1 | 1 | 1 | 1 | 1 | 1 | 1 | 1 | 1 | 1 |
| Completed primary | 0.16 (0.00 to 9.09) | [38.65] (0.13 to 11919.09) | 0.21 (0.03 to 1.38) | 0.17 (0.02 to 1.65) | 0.76 (0.17 to 3.48) | 0.52* (0.28 to 0.98) | 0.1* (0.01 to 0.82) | 2.18 (0.29 to 16.55) | 0.13 (0.02 to 1.02) | 0.95 (0.23 to 3.92) | 0.21 (0.03 to 1.68) | 0.1* (0.01 to 0.89) | NA | 0.56 (0.08 to 4.04) |
| Completed secondary | 0.59 (0.00 to 113.42) | NA | 0.23 (0.04 to 1.29) | 0.44 (0.09 to 2.25) | 0.78 (0.2 to 3.08) | 0.68* (0.5 to 0.92) | 0.65 (0.36 to 1.15) | 2.94 (0.4 to 21.91) | 0.24 (0.04 to 1.38) | 0.58 (0.19 to 1.82) | 1.38 (0.12 to 15.99) | 0.06* (0 to 0.71) | 0.75 (0.41 to 1.38) | 0.33 (0.0 to 1.19) |
| Completed college/university | 0.41 (0.01 to 17.31) | [67.97] (0.65 to 7054.79) | 0.24 (0.05 to 1.2) | 1.08 (0.21 to 5.52) | 1.28 (0.31 to 5.29) | NA | NA | 1.31 (0.16 to 10.51) | 0.39 (0.06 to 2.32) | NA | 1.21 (0.12 to 12.55) | 0.03* (0 to 0.73) | NA | NA |
| **Occupation** | | | | | | | | | | | | | | |
| Non-employed# | 1 | 1 | 1 | 1 | 1 | 1 | 1 | 1 | 1 | 1 | 1 | 1 | 1 | 1 |
| Employed | 9.16 (0.29 to 284.89) | 16.73 (0.28 to 999.11) | 1.01 (0.49 to 2.09) | 1.8 (0.84 to 3.84) | 0.96 (0.41 to 2.26) | 1 (0.71 to 1.41) | 1.64 (0.94 to 2.88) | 0.69 (0.4 to 1.19) | 0.58 (0.28 to 1.2) | 0.41 (0.11 to 1.49) | 3.89 (0.44 to 34.75) | 6.65 (0.49 to 89.96) | 0.95 (0.43 to 2.08) | NA |
| **Wealth index** | | | | | | | | | | | | | | |

**Table 4** Continued

| Predictors | African region | | Eastern Mediterranean | European region | | | | Regions of the Americas | | South-East Asia region | | Western Pacific region | | |
|---|---|---|---|---|---|---|---|---|---|---|---|---|---|---|
| | Ethiopia | Senegal | Qatar | Greece | Kazakhstan | Russian Federation | Ukraine | Costa Rica | Mexico | Indonesia | India | Malaysia | Philippines | Vietnam |
| | AOR (95% CI) | AOR (95% CI) | AOR (95% CI) | AOR (95% CI) | AOR (95% CI) | AOR (95% CI) | AOR (95% CI) | AOR (95% CI) | AOR (95% CI) | AOR (95% CI) | AOR (95% CI) | AOR (95% CI) | AOR (95% CI) | AOR (95% CI) |
| Rich# | 1 | 1 | 1 | 1 | 1 | 1 | 1 | 1 | 1 | 1 | 1 | 1 | 1 | 1 |
| Middle | 0.08 (0 to 2.42) | NA | NA | NA | 0.68 (0.26 to 1.8) | 0.57* (0.35 to 0.94) | 1.46 (0.77 to 2.75) | 0.85 (0.37 to 1.94) | 1.02 (0.48 to 2.19) | 1.88 (0.47 to 7.57) | 1.29 (0.15 to 11.28) | 0.92 (0.27 to 3.17) | 0.52 (0.23 to 1.17) | 0.71 (0.17 to 3.02) |
| Poor | [58.64]*** (6.99 to 491.99) | NA | NA | 0.73 (0.26 to 2.06) | 0.86 (0.41 to 1.81) | 0.95 (0.69 to 1.31) | 0.54 (0.24 to 1.21) | 0.62 (0.32 to 1.2) | 0.17** (0.05 to 0.54) | 0.47 (0.05 to 4.19) | 4.66 (0.28 to 76.22) | 0.29 (0.01 to 6.79) | 0.42 (0.16 to 1.08) | 0.76 (0.08 to 7.49) |
| Currently smoke tobacco | | | | | | | | | | | | | | |
| No# | 1 | 1 | 1 | 1 | 1 | 1 | 1 | 1 | 1 | 1 | 1 | 1 | 1 | 1 |
| Yes | 2.38 (0.08 to 68.27) | 2.93 (0.22 to 38.99) | [17.68]*** (6.56 to 47.67) | 2.26* (1.01 to 5.04) | [16.97]*** (7.46 to 38.59) | 5.60*** (4.15 to 7.54) | 3.62*** (2.04 to 6.41) | 6.66*** (3.83 to 11.59) | 4.8*** (2.36 to 9.76) | 5.33 (0.99 to 28.55) | 4.86* (1.12 to 21.14) | 13.42* (1.31 to 137.7) | 12.19*** (5.27 to 28.24) | [458.5]*** (27.08 to 7763.37) |
| Advertisements in stores | | | | | | | | | | | | | | |
| No# | 1 | 1 | 1 | 1 | 1 | 1 | 1 | 1 | 1 | 1 | 1 | 1 | 1 | 1 |
| Yes | NA | 17.29 (1.5 to 198.96) | 1.45 (0.63 to 3.36) | 1.82 (0.92 to 3.61) | 0.99 (0.45 to 2.2) | 1.32 (0.83 to 2.09) | 1.71 (0.93 to 3.16) | 0.63 (0.23 to 1.67) | 1.18 (0.55 to 2.54) | 1.3 (0.55 to 3.07) | 0.97 (0.22 to 4.25) | 0.78 (0.17 to 3.65) | 0.86 (0.48 to 1.54) | 1.76 (0.4 to 7.74) |
| Advertisements on internet | | | | | | | | | | | | | | |
| No# | 1 | 1 | 1 | 1 | 1 | 1 | 1 | 1 | 1 | 1 | 1 | 1 | 1 | 1 |
| Yes | NA | NA | 1.07 (0.36 to 3.17) | 1.63 (0.44 to 6.13) | 0.63 (0.22 to 1.74) | 1.41 (0.96 to 2.07) | 0.61 (0.23 to 1.63) | 1.14 (0.31 to 4.28) | 1.18 (0.51 to 2.74) | NA | 0.72 (0.05 to 10.21) | 0.46 (0.05 to 3.88) | 2.96** (1.41 to 6.21) | 2.05 (0.1 to 41.01) |

***p<0.001, **p<0.01, *p<0.05.
[] AOR is typically high as the odds of e-cigarette use in the reference population is too low compared the predictor population and thus not interpretable in general.
NA denotes data not available or cases not available.
#Reference category.
AOR, Adjusted Odds Ratio.

**Table 5** Estimates of country level average prevalence of e-cigarette use and the total number of users, Global Adult Tobacco Survey, 2011–2017

| WHO region | Country | Survey year | Total population* | Average prevalence (predicted) |
|---|---|---|---|---|
| African region | Ethiopia | 2016 | 103 603 501 | 0.016 |
| | Senegal | 2017 | 15 419 381 | 0.007 |
| Eastern Mediterranean | Qatar | 2013 | 2 336 574 | 0.018 |
| European region | Greece | 2013 | 10 965 211 | 0.019 |
| | Kazakhstan | 2014 | 17 288 285 | 0.033 |
| | Russian Federation | 2016 | 144 342 396 | 0.037 |
| | Ukraine | 2017 | 44 831 135 | 0.030 |
| Region of the Americas | Costa Rica | 2015 | 4 847 804 | 0.024 |
| | Mexico | 2015 | 124 777 324 | 0.017 |
| South-East Asia region | Indonesia | 2011 | 245 116 206 | 0.029 |
| | India† | 2017 | 1 338 658 835 | 0.012 |
| Western Pacific region | Malaysia | 2011 | 28 650 955 | 0.039 |
| | Philippines | 2015 | 102 113 212 | 0.022 |
| | Vietnam | 2015 | 92 677 076 | 0.012 |

*https://data.worldbank.org/indicator/SP.POP.TOTL?most_recent_year_desc=false.
†Reference category.

2017 had a prevalence of 33 and 30 persons per thousand population of e-cigarette use (table 5).

## DISCUSSION

The present study has examined the awareness of e-cigarette use, prevalence and socio-demographic determinants using the most recent rounds of the GATS covering 14 selected countries. The developed nations like USA, UK, New Zealand, France and Germany have their own individual surveys to provide general/population-specific estimates of e-cigarette use and awareness. It was estimated that the prevalence of e-cigarette use among US middle and high school students increased from 3% in 2011 to 7% in 2012.[41] According to the Local Tobacco Control Profiles for England annual report, there had been a significant increase in e-cigarette users (population aged ≥16) from 15% in 2014 to 19% in 2017 in UK. It had also been found that use of e-cigarette was much more prevalent among the younger population (aged 16–24 years) than the older population (aged ≥60 years).[42] Quatremère et al[43] estimated that 4% of the population in Mainland France were e-cigarette users and 3% of them used it on a daily basis. Of these total users, 60% were men, and 58% had a bachelor's degree. Eichler et al[44] using a population-based cross-sectional survey, conducted during 2016, found that 49% of males and 51% of females used e-cigarettes in Germany. In New Zealand, 7% of the general population used e-cigarettes, with young persons (18–24 years) were more likely than those over 45 years.[45] Furthermore, independent of educational qualification, persons with higher income were more likely to use e-cigarettes than those with a moderate or lower income in New Zealand.[45]

This study suggests that countries in the European and Eastern Mediterranean regions have a higher awareness of e-cigarette than the countries of other regions. The awareness and current use of e-cigarettes are higher among urban population in almost all the countries studied. On the contrary, Greece and Vietnam show higher prevalence of awareness in the rural areas. It is possible that the exposure to advertisement, accessibility and availability of e-cigarettes may have led to higher e-cigarette use in urban areas.[46]

The prevalence of current e-cigarette use is higher in Russian Federation (4.39%) because e-cigarettes are not covered under the tobacco control policy of Federal Law No. 15-FZ.[47] Therefore, the restriction on marketing and advertisement finds a policy gap.[33] Though e-cigarette regulation policies in Malaysia (3.94%) and in Ukraine (3.28%), puts a strict restrictions on distribution, importation, minimum age and in sales; yet the prevalence of e-cigarette use is moderately high in these two countries.[48] On the contrary, the least prevalence of e-cigarette use in the countries like India (0.66%) and Vietnam (1.88%) is due to the ban on manufacturing, imports, sales, advertisement, and in distribution.[48–50]

The study suggests that the urban population, males, young people (15–24), those with higher wealth scores and the higher educated individuals keep better knowledge of e-cigarettes across all the selected countries. This study has shown that e-cigarette use among the younger adults is high compared with the older adults across all the selected countries; which emphasises the need of

implementing new population specific policies to curb the use of nicotine at younger ages. Notably, the cross-country analysis does not show consistent association between use of e-cigarette and educational qualification across selected GATS countries. While the wealthy population is more likely to use e-cigarettes.

Study on women and e-cigarette use is less. This study shows that males have higher prevalence of e-cigarette use than the females, except in Greece. A study by Tzortzi et al., 2020 reported that both tobacco smoking and e-cigarette use among females are higher than males in Greece, which is attributable to better economic condition. [51] The gender gap in e-cigarette awareness in wide in Greece, post 50 years of age, while the gap is distinct in early ages in countries like Kazakhstan and Qatar . The gender gap in use of e-cigarette is negligible in most of the countries except among the younger cohorts of Russia, Philippines Malaysia and Indonesia where prevalence of use is distinctly more among the younger males. A study based on an online survey in USA, had found that male older adults use e-cigarettes as a medium to quit smoking while the initiation of e-cigarette use among females is influenced by their family and friends. [52] The multivariate adjusted model of this study also reveals that males have higher use of e-cigarette which could be due to quitting tobacco smoking. In the counterpart, women's initiation of e-cigarette use is often restricted by social stigma and health concerns. [53] On the other hand, males purchase e-cigarettes independently while females rely on their peers, limiting their use. [54 55] Literature says that all current female e-cigarette users were dual tobacco users, including pregnant women. The frequent reasons among all ever users were that they wanted to quit smoking, thought it would be less expensive, could use e-cigarettes where smoking is prohibited, and thought they would be less harmful. It is necessary to determine the impact of e-cigarette use on maternal and infant health. [56]

In Costa Rica and Mexico, young adults use e-cigarettes, suggesting a relatively higher availability, affordability and moderate restrictive policies on nicotine vaping products.[57] Previous studies on tobacco smoking suggest that urban youths are more likely to use tobacco than their counterparts which may be due to urban accessibility, urban space and advertisements.[34 58] Literatures also suggest that early employment is associated with daily use among youths.[59] Adolescents/youths who are either employed or receive more pocket money and supposedly enjoy economic independence are more likely to smoke and use other addictive substances.[59 60] The present study shows that the unemployed population has a lower prevalence of e-cigarette use, possibly due to a lack of discretionary income.[61 62]

The findings of this research are also in line with other studies explaining the association of the advertisement/promotions through various media that trigger the use of e-cigarette among young adults.[54 56 63] The use of the internet and social media for advertising, marketing and ENDS promotion has expanded rapidly and concerns

have been raised regarding deceptive health statements, claims on withdrawal effectiveness, and hence targeting the youths.[64] Although the ill-health effects of e-cigarette use are debated, it is still established that the aerosols of the majority of ENDS contain toxic chemicals that are hazardous to lung health and elevate the risk of cardiovascular diseases.[65 66] The promotion and marketing challenges of e-cigarettes revolve around the country's public experiences and tobacco control measures. In many countries, such as Australia, Brazil, Denmark and others, e-cigarettes advertising and marketing are illegal.[48] On the other hand, sales in other countries are legal and authorised.[67] Interestingly, in UK, despite a strong tobacco policy, e-cigarettes use is booming.[14] At the same time, countries like Costa Rica, Ecuador, Honduras, the Republic of Korea, Togo and Vietnam do not have regulations on sales of e-cigarettes.[48 67] Initially, the e-cigarettes manufacturers were independent and then transgressional tobacco companies came to venture into the market by creating a new form of nicotine consumers (young and non-smokers) pretending to curtail the smoking epidemic as a reputational tactic.[68] Although the present study is cross-sectional, it fails to capture the association of purchasing behaviour and employment. Therefore, it generalises the likelihood of economic stability and e-cigarette use across selected GATS countries.

India has banned electronic cigarettes in 2019 considering its health side effects and addictive properties. The study indicates very low awareness and use of these products in India. Thus, the country is not in the verge of e-cigarette addiction as it is established that higher awareness is strongly correlated with a trial of e-cigaarette. However, point to ponder is the higher prevalence of e-cigarette use among females over males in age 50s and 60s. There is a surge of contradictory arguments in favour and against this ban. The reasons cited for banning e-cigarette are damage in DNA, carcinogenesis, cellular, molecular and immunological toxicity, respiratory, cardiovascular and neurological disorders and adverse impact on foetal development and pregnancy, risk of initiation of tobacco to non-smokers leading to opening a gateway for new tobacco addiction. Unfortunately, anti-ban school says that these decisions were made without previously examining the patterns of e-cigarette use in India and the profile, smoking status, and perceived benefits or harms among local users. A recent study in India states that e-cigarette is an alternative nicotine delivery system with significantly less harmful emissions than smoke, and thus could be an option for those who are unable to achieve smoking abstinence using other available means.[69] However, this study does not reveal such strong association with age, i.e older cohort using e-cigarette more. Based on large scale youth survey, it is observed that most restrictive policies such as the ban on e-cigarettes appear to reduce e-cigarette use among the youth.[70]

An important strength of this study is that it provides the baseline e-cigarette prevalence for multiple countries. As GATS data collections continue, the prevalence of

e-cigarette use in each country can continue to be monitored and can be used to evaluate policy changes that occur. However, this study has some limitations. First, we have considered only 14 countries in the temporal setting of 2011–2017. Thereafter, we could not incorporate the drastic changes in e-cigarette devices, brands, marketing between 2011 and 2017, which might have impacted the prevalence of e-cigarette use. Finally, the outcome variables in this particular study are self-reported in nature, thus the study estimates are limited to self-reporting bias and system bias. The self-reporting bias among the individuals largely depends on their socio-economic and demographic characteristics, which has been controlled in the study. To address the system bias in the estimations, we used the country-specific survey weights to derive the estimates. Notably, the GATSs are popularly referred and used across countries to provide the prevalence of tobacco and e-cigarette use in the surveyed population using the self-reported information only. At the same time, according to the WHO-GATS protocol, the survey instruments were developed to collect the necessary information on e-cigarette awareness and its use and thus we are restricted to use these self-reported information in this study in the absence of any other relevant data for these countries. Nevertheless, despite a few lacunae, there are gaps in the knowledge of cross-country e-cigarette use and this study provides useful clues to policymakers for establishing effective measures for public health planning.

## CONCLUSION

With a cross-country analysis of GATS datasets, this study presents the most recent socio-economic and demographic patterns of e-cigarettes awareness, use and its determinants. The study further estimates the predicted prevalence of e-cigaratte use of 14 countries. The findings from this study show that e-cigarette use is varied across specific sub groups of selected GATS countries, with a higher usage among males, youth aged 15-24 and among urban population. Country specific advertisement in promoting e- cigaratte plays crucial role in higher usage of e- cigaratte. Two countries that deviate from the usual trend are Greece and Vietnam where e-cigeratte smoking is more among females and in rural population respectively. The gender gap in use of e-cigarette is negligible in most of the countries except among the younger cohorts of Russia, Philippines, Malaysia and Indonesia where prevalence of use is distinctly more among the young males. On the contrary, relatively higher prevalence of e-cigarette smoking among females in the older adult age observed in Russia and Ukraine along with some of the Asian countries, i.e. India, Philippines, Qatar. However, relatively higher prevalence of e-cigarette smoking among females in the older adult age observed in some of the Asian countries, i.e. India, Philippines, Qatar needs further research. Better education and wealth do guarantee better awareness but do not reveal any strong association with usage.

Russia, Ukraine, Costa Rica and Mexico need detailed study to explore whether e-cigaratte use is an indulgence to new mode of addiction, youth being highly likely to adopt this practice. While Indonesia, Vietnam, Philippines and India must investigate whether e-cigarette is an option for regular tobacco control across ages, age being an insignificant predictor of use. Further, e-cigarette use being more among smokers and among urban population, area specific surveillance is needed to study whether this practice is actually helping people to quit tobacco smoking or pushing youth to pick up the regular tobacco smoking behaviour over time.

**Author affiliations**
[1]Department of Survey Research & Data Analytics, International Institute for Population Sciences, Mumbai, Maharashtra, India
[2]Centre of Social Medicine and Community Health, Jawaharlal Nehru University, New Delhi, Delhi, India
[3]Department of Population and Development, International Institute for Population Sciences, Mumbai, Maharshtra, India
[4]Department of Population and Development & Center of Demography of Gender, International Institute for Population Sciences, Mumbai, Maharashtra, India
[5]Center of Demographic, Urban and Environmental Studies, El Colegio de Mexico A.C, Mexico City, Mexico
[6]Department of Population Policies & Programmes, International Institute for Population Sciences, Mumbai, Maharashtra, India

**Acknowledgements** Authors are immensely thankful to the Bill and Melinda Gates Foundation's grant, BMGF INV-047356, for providing APC support. This grant helps innovative research under the Center of Demography of Gender (CDG), International Institute for Population Sciences, Mumbai. This research work was first accepted by the BMJ-open on 13th September 2021, and we had to withdraw the article on 21st January 2022 due to failure of APC payment. After 2 years of long wait, we are seeing this work getting published and thus expressing our gratitude to the funder, BMGF and the journal BMJ Open.

**Contributors** SK, SS, and JK: conceptualisation, methodology, validation, formal analysis, investigation, writing (original draft preparation). AC: conceptualisation, methodology, validation, investigation, writing (original draft preparation) and editing. EAB: editing and review BP: validation. All authors have read and approved the final version of the manuscript. SK is responsible for overall content as guarantor

**Funding** The funder paid the article processing charges

**Competing interests** None declared.

**Patient and public involvement** Patients and/or the public were not involved in the design, or conduct, or reporting, or dissemination plans of this research.

**Patient consent for publication** Not applicable.

**Ethics approval** Not applicable.

**Provenance and peer review** Not commissioned; externally peer reviewed.

**Data availability statement** Data are available upon reasonable request. The datasets used and/or analysed during the current study are available publicly and also from the corresponding author on reasonable request (online supplemental table 1).

**ORCID iDs**
Sampurna Kundu http://orcid.org/0000-0002-6386-9040
Subhojit Shaw http://orcid.org/0000-0003-2612-3175
Junaid Khan http://orcid.org/0000-0003-4662-2318
Aparajita Chattopadhyay http://orcid.org/0000-0002-1722-4268
Emerson Augusto Baptista http://orcid.org/0000-0001-7582-2736

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
