## [Reviewer comments · BMJ Open]

ARTICLE DETAILS

TITLE (PROVISIONAL)	Awareness and determinants of e-cigarette use across selected WHO region countries: Evidence from the Global Adult Tobacco Survey (GATS)
AUTHORS	Kundu, Sampurna; Shaw, Subhojit; Khan, Junaid; Chattopadhyay, Aparajita; Baptista, Emerson; Paswan, Balram

VERSION 1 – REVIEW

REVIEWER	Michael Blaha Johns Hopkins Hospital, Baltimore, USA, Medicine/Cardiology
REVIEW RETURNED	18-Mar-2021

GENERAL COMMENTS	In this study, Kundu and colleagues assessed awareness and use of e-cigarettes in 14 countries, and examined the sociodemographic characteristics associated with use. The authors report that of the 14 countries studied, e-cigarette awareness and prevalence of use were highest in Russia, and lowest in India. Also, sex, age, place of residence, occupation, and combustible cigarette use were significant predictors of e-cigarette use in most countries. It is a timely topic considering the high uptake of e-cigarettes in many societies, especially among younger individuals. I have attached a few comments that will hopefully further strengthen the manuscript. General comment Kindly review the manuscript to check for and fix grammatical errors. The authors should also consider reducing the number of repeated sentences to make for a more concise and impactful publication. Lastly, many of the cited references are not primary sources. The authors should consider citing more of the original studies to allow for verification of references by interested readers. Abstract Line 23: It appears you are assessing “awareness about e-cigarettes” not “awareness of e-cigarette use”. Also consider rephrasing “patterns of e-cigarette use prevalence” to “e-cigarette use prevalence and patterns”. Line 30: Kindly define “GATS” at first mention Line 31: consider replacing “e-cig” with “e-cigarette use”
--

Line 33: what do you mean “part of the post-estimation”? the sentence seems to be missing words.

Line 41: “subject to awareness” does not tie in with the rest of the sentence.

Introduction:

General comment: Consider shortening this section to a page or less and making the language more speculative as it relates to e-cigarettes, as many of the definitively stated comments are either unproven or have been countered in other papers not cited by the authors.

Page 4:

Line 9: Please cite the source of the stated prevalence estimate “3 million US adolescents”.

Line 12: consider deleting “the following components”

Line 25: Consider rephrasing “e-cigarettes have the potential to be a gateway for quitting smoking”. E-cigarettes are either considered a "gateway for subsequent cigarette smoking" or "a form of tobacco cessation therapy"

Lines 26-33: Can be summarized into one sentence.

Line 27: “On the other hand, studies have found that the odds of quitting cigarettes were lower for those who had not formerly smoked” This phrase is confusing, please review. Also, the cited study examined trends in e-cigarette and combustible cigarette use among students in Poland and found that prevalence increased for both. The study was not about quitting smoking, kindly review the reference.

Line 33: the cited study was conducted among adolescents, not young adults. Kindly review the reference.

Line 39: “Although the aerosols from the e-cigarettes are not harmful...” E-cigarette aerosols might contain toxic chemicals and might be harmful. Consider adjusting language.

Page 5:

Lines 16-37: Consider shortening or eliminating this portion of the introduction.

Line 42: “The present study offers a means for the countries involved to inform themselves about young adults’ initialization of e-cigarettes use”. The cited study does not provide information on young adults’ initiation of e-cigarette use, kindly review.

Data:

Page 6:

Line 31: Kindly define "PPS" at first mention.

Table 1: Kindly define "HHs" in footnote

Statistical Analyses(Page 7):

Line 25: "further we have used bivariate and multivariate analyses"
This sentence appears incomplete. Kindly revise.

Line 30: Age was not presented in 10-year groups in any of your tables, please revise.

The survey years are different for the various countries e.g. 2011 in Indonesia and 2017 in Ukraine. How do you account for this when making comparisons among countries? This is a critical point that must be addressed to make sense of this paper.

Since survey data were used in this study, did you apply weights to make estimates nationally representative? If so, please state.

Also, what statistical test did you use to examine the difference in e-cigarette awareness or use among the different comparison groups?

What p-value determined statistical significance?

Results:

Page 8:

Line 14: kindly include an age group for "younger people".

Line 36: "though the association was not significant"... This is confusing. Consider writing out what association you are referring to

Line 41: "...had high usage among the younger age groups". The prevalence rates reported to support this vary widely. The authors also state in line 46-47 that usage was "high" among older age groups, and report prevalence rates as low as 1.25%. Perhaps state what you consider "high" or consider using a comparator.

Line 44: from the table, residence was also significant.

Page 9:

Line: 10: Since you used multiple independent variables and one dependent variable (outcome), multivariable is the appropriate term to use.

Line 18: Consider replacing "maximum" with "highest"

Page 10:

Table 2: Does "NA" imply that variable is missing for that specific country? For instance, there was no data for females in Russia.

	Does this mean survey was limited to only males? If so, please provide this information under the Data Section. Page 11: Table 3: Please define what “****” and “*” mean. Also, please format the total population and projected users appropriately. Discussion: Page 12: Line 22: Although in Greece, females had higher prevalence of e-cigarette use than males, this was not significant and so I do not think you can state that females were more likely than males to use e-cigarettes. Line 27: Is this a reason why more females use e-cigarettes than males in Greece? Line 32: “This raises the seriousness to facilitate initiation of nicotine use at the early stage of life” Not clear, please review Line 38: The cited reference [2] does not state this. Kindly review. Line 40: The cited reference [39] does not state this. Kindly review. Line 44: “GATS being a cross-sectional survey, we do know whether a former smoker who has successfully given up smoking did so by using e-cigarettes as a cessation method” Please revise. Line 56: This citation [41] is not needed since you are only stating your results. Page 13: Lines 3-6: This has already been stated on page 12 lines line 24-24. Line 9: “Among countries with higher incomes, such as Costa Rica and Mexico, young adults are more exposed, which may suggest that ecigs users are affected by friends”. How does use in young adults suggest that users are affected by friends? Line 37: “Secondly, GATS being the cross-sectional survey we could not able to identify whether the current e-cigarettes user has quit the use before initiation” This is confusing, please revise. Conclusion The authors should consider shortening this section.
--	---

REVIEWER	Adam Cole University of Ontario Institute of Technology
REVIEW RETURNED	06-Apr-2021

GENERAL COMMENTS	Comments for the authors: This study identifies the prevalence of awareness and use of e-
--

cigarettes among adults in 14 countries. Most surveillance research to-date has focused on developed countries (e.g., US, UK, Canada) and there are few data highlighting the prevalence of e-cigarette use in other countries. The data are relevant for an international audience and can help to inform and evaluate e-cigarette policies. However, I believe there are major edits that are necessary before the manuscript is adequate for publishing. I believe the manuscript would benefit from English language review and editing, and the Methods require additional details. The presentation of the Results could be simplified so they are more easily understood and interpretable.

Major comments

-Abstract (page 3, line 11): This statement is incorrect since the analyses did not identify the likelihood of e-cigarette use among different subgroups or characteristics. This would require logistic regression analyses and presenting the results (ORs) from the models.

-Abstract (page 3, lines 12-16): I'm not sure I understand these points. What cessation behaviour is being referred to – cigarette smoking or e-cigarette use? Since the data are cross-sectional, both behaviours are being measured at the same point in time, so it doesn't seem relevant whether or not the individual quit smoking before using e-cigarettes.

-Introduction (page 4, lines 23-53): The second paragraph of the introduction seems to focus on the use of e-cigarettes for smoking cessation. There should be some discussion about the association between e-cigarette use and smoking initiation, particularly among young people given the age range of the sample. Any e-cigarette policy would need to balance the risks of use among young people with the benefits of encouraging smokers to quit.

-Introduction (page 5, lines 46-51): The authors indicate that IGTC has measured e-cigarette advertising and promotion since 2014. Why haven't the authors included any of these data in their analyses? It would seem these policy data combined with e-cigarette use data from GATS would help to evaluate and inform e-cigarette regulation across countries.

-Introduction (page 6, lines 8-18): It seems like a better rationale for the current study is the need for e-cigarette awareness and prevalence data from countries other than the US and the UK (where most current evidence is from). Cross-country data are relevant for evaluating different policy environments.

-Statistical analysis (page 7, lines 35-39): How were place of residence, wealth index, tobacco users, and advertisement source determined? Do tobacco users include cigarette smokers, smokeless tobacco users, and bidi users? What does "advertisement source" mean? More explanation of how these variables were derived is needed.

-Statistical analysis (page 7): I believe the GATS includes sample weights to account for the sampling design. Did the authors use these when calculating country-specific prevalence rates? This would seem to be a more robust method of calculating country-level prevalence rates.

-Statistical analysis (page 7, lines 39-40): The authors indicate they calculated country-wide adjusted prevalence rates of current e-cigarette use. How was this done? More explanation is needed to

assess whether this is appropriate.

- Statistical analysis (page 7): I suggest adding calculations that show the ratio of those who currently used e-cigarettes given they have heard about them [see Zhu et al. (2021). Early adoption of heated tobacco products resembles that of e-cigarettes. *Tobacco Control*. <https://doi.org/10.1136/tobaccocontrol-2020-056089>; or Zhu et al. (2013). The use and perception of electronic cigarettes and snus among the U.S. population. *PLoS One*, 8(10), e79332. <https://doi.org/10.1371/journal.pone.0079332>]. These ratios can be compared across countries to show differences in the adoption of e-cigarettes. Despite high awareness, relatively few people in Greece or the Russian Federation report current e-cigarette use, compared with low awareness and low prevalence in India or Ethiopia. The results of this calculation could be added as a new table or described in the text.
- Figure 1: Given that GATS includes sample weights and the survey is nationally representative, 95% confidence intervals should be added to the figure estimates.
- Supplementary Table 1: I recommend that the supplementary table be included as part of the manuscript, especially since the results of this table are described in the main text and it is the only place where awareness data are presented.
- Results (page 9, lines 12-16): The authors indicate that the estimates were adjusted. How was this done? It hasn't been adequately described in the Statistical Analysis.
- Table 2: Rather than showing the estimates per 1000 population, it makes more sense to show a prevalence rate (%) since this is easier to interpret and compare. This table also does not show a predicted prevalence, it just shows the prevalence of e-cigarette use at the time of the survey according to sociodemographic characteristics (nothing is being predicted). Changes to this table should be reflected in the results on page 9 (lines 18-34).
- Table 2: I suggest splitting the youngest age group into 15-18 (youth) and 19-24 (young adults) given concerns about the increasing popularity of e-cigarettes among underage youth.
- Table 2: The age categories do not align with those described in the methods. Why were these age categories chosen?
- Table 2: Why aren't education level, wealth index, and advertisement exposure included in this table? Given the discussion of the importance of advertising in the introduction, it seems like this should be included in the table.
- Results (page 11, lines 15-23): Again, the authors indicate that the estimates were adjusted, but have not described how this was done.
- Table 3: I'm not sure what value this table adds to the manuscript outside of the adjusted odds ratio (but even that is limited). I suggest removing this table.
- Table 3: I'm not sure why the projected number of total users is included in this table or how it was calculated. It's not actually a projected number of users, but rather an estimated number of users. I suggest removing it as it does not add a lot of value to the manuscript. Any changes should be reflected in the results on page 11 (lines 50-60).
- Discussion (page 12, line 13): From my review of the manuscript, the analyses did not identify predictors of e-cigarette use across country, so this statement is incorrect and should be removed.
- Discussion (page 12, lines 53-54): Again, from my review of the

	manuscript, the analyses did not identify associations between sociodemographic characteristics and e-cigarette use across countries. The analyses were basically descriptive and did not measure associations. Therefore this statement is incorrect and should be removed. -Discussion (page 13, line 35): A major limitation is the range of years included. There have been drastic changes in e-cigarette devices between 2011 and 2017 that need to be acknowledged as they have impacted the brands of e-cigarettes, e-cigarette marketing, and the prevalence of e-cigarette use. -Conclusion (page 13, line 57): Again, the analyses did not measure associations, so statements about “predictors of e-cigarette use” should be removed. Minor comments -Introduction (page 4, lines 25-28): The authors indicate that multiple studies suggest e-cigarettes are a gateway for quitting smoking/prevent people from quitting smoking, but only one reference is provided. I suggest adding in a couple more references (or even a review article) to support your points. -Introduction (page 4, lines 39-44): The discussion about marijuana in e-cigarettes is distracting and irrelevant to the current study. I suggest the authors remove it. -Introduction (page 4, line 48): I’m not sure what the authors are trying to say here: “The efficacy to quit smoking has relieved the dependency on nicotine.” -Introduction (page 5, line 21): I’m not sure what the authors mean by “astatically”. Data (page 6, line 31): What does “PPS” stand for? This acronym should be spelled out for the reader. -Figure 1: The axis should be “Percentage of adults >15 years” (or something similar) since the graph shows both awareness and prevalence of current e-cigarette use -Results (page 8, lines 27-32): I suggest adding a reference to Figure 1, as it was unclear where these prevalence rates could be found in the manuscript. Results (page 8, line 56): The authors indicate that current e-cigarette use is higher for current users than for non-users. Users of what product? Table 2: Are the current tobacco smoker results for Mexico correct? It currently shows that e-cigarette use is higher among non-current smokers than current smokers. Discussion (page 12, line 44): The line should read “GATS being a cross-sectional survey, we do not know...”
--	---

REVIEWER	Allison Glasser New York University School of Global Public Health
REVIEW RETURNED	10-Apr-2021

GENERAL COMMENTS	Overall: This study is one of the first to compare prevalence of awareness of e-cigarettes and e-cigarette use across countries in the WHO regions. Prior to being considered for publication, the manuscript should be edited for English grammar and vocabulary, as there are numerous places throughout the manuscript where the meaning of
---

the sentence(s) was unclear. In addition, the Introduction and Discussion sections need some close editing as per my recommendations below.

Strengths/Limitations:

The 4th bullet (“GATS being a cross-sectional survey, we could not identify whether the current ecigarettes user has quit the use before initiation.”) is difficult to understand. Consider rewording.

Introduction:

Please provide a citation for the 2nd sentence in the Intro.

The sentence in lines 26-28 (page 4) does not make sense: “On the other hand, studies have found that the odds of quitting cigarettes were lower for those who had not formerly smoked.”

Lines 28-33, page 4: There are two issues with this statement. First, it is not true – vaping among adult never smokers has never exceeded vaping among current smokers in the US (most recent pub: Margaret Mayer; Carolyn Reyes-Guzman; Rachel Grana; et al Kelvin Choi; Neal D. Freedman. Demographic Characteristics, Cigarette Smoking, and e-Cigarette Use Among US Adults. JAMA Netw Open. 2020;3(10):e2020694.

doi:10.1001/jamanetworkopen.2020.20694). Second, the citation at the end of the sentence (citation 9) is a study conducted among adolescents, not adults, and is longitudinal, not cross-sectional.

The sentence in lines 34-37, page 4 (“These chemicals have adverse effects on children, adolescents, and pregnant mothers, which may contribute to cardiovascular disease.”) needs a citation.

Line 48 (page 4): I don’t understand this sentence, “The efficacy to quit smoking has relived the dependency on nicotine.”

I also do not understand the sentence on lines 21-24 on page 5, “Evidently a study conducted by Pepper et al. [21] found that relatively higher advantageous substitute (e-cigarettes) has higher rate of adaptation.”

Starting with line 45 of page 5 (“A handful of studies...”), the remainder of this paragraph is not cohesive nor does it fit with the first half of the paragraph. The statements jump from marketing, to taxes, to cessation, and it is not clear what the point of this section is.

Results:

At the bottom of page 8, do “users” and “non-users” refer to cigarette smokers? If it refers to e-cigarette use, then the sentence does not make sense.

Discussion:

I recommend the authors describe the main findings across the bivariate and multivariate models instead of separately, as it feels there is a lot of repetition.

There is no discussion of cross-country differences in prevalence in

	the Discussion section. Could differences be related to policy environments, stage in the tobacco epidemic, etc.?
--	---

VERSION 1 – AUTHOR RESPONSE

Reviewer 1

Dr. Michael Blaha, Johns Hopkins Hospital, Baltimore, USA

In this study, Kundu and colleagues assessed awareness and use of e-cigarettes in 14 countries, and examined the sociodemographic characteristics associated with use. The authors report that of the 14 countries studied, e-cigarette awareness and prevalence of use were highest in Russia, and lowest in India. Also, sex, age, place of residence, occupation, and combustible cigarette use were significant predictors of e-cigarette use in most countries.

It is a timely topic considering the high uptake of e-cigarettes in many societies, especially among younger individuals. I have attached a few comments that will hopefully further strengthen the manuscript.

A: We appreciate the comments, suggestions, and careful reading. We have tried to incorporate the suggestions as best as possible and explain the doubts more clearly. We believe that the current version of the manuscript has improved greatly based on your comments and suggestions.

Kindly review the manuscript to check for and fix grammatical errors. The authors should also consider reducing the number of repeated sentences to make for a more concise and impactful publication. Lastly, many of the cited references are not primary sources. The authors should consider citing more of the original studies to allow for verification of references by interested readers.

A: Thank you for the suggestions. We have reviewed the issues you raised.

Abstract

Line 23: It appears you are assessing “awareness about e-cigarettes” not “awareness of e-cigarette use”. Also consider rephrasing “patterns of e-cigarette use prevalence” to “e-cigarette use prevalence and patterns”.

A: Thank you for the suggestion. We have modified the sentence.

Line 30: Kindly define “GATS” at first mention

A: Thank you for the comment. We have defined GATS as “Global Adult Tobacco Survey” at the first place.

Line 31: consider replacing “e-cig” with “e-cigarette use”

A: Thank you for the comment. We have replaced the word “e-cig” with “e-cigarette use” throughout the manuscript.

Line 33: what do you mean “part of the post-estimation”? the sentence seems to be missing words.

A: Thank you for the comment. We have modified the method section in the abstract

“We selected 14 countries from six different WHO regions where Global Adult Tobacco Survey (GATS) was conducted in different years during 2011-2017. Using bivariate cross-tabulation, we estimated the prevalence of e-cigarette use by socio-economic and demographic characteristics. In addition, country specific multivariable logistic regression was estimated to examine the pattern and determinants of e-cigarette use in the respective countries.”

Line 41: “subject to awareness” does not tie in with the rest of the sentence.

A: Thank you for the comment. We have modified the result section in the abstract

“Among the selected GATS countries, awareness of e-cigarette was highest in Greece (88%), whereas it was observed to be lowest in India (3%). On the other hand, the prevalence of e-cigarette use was highest in Russia (4%) and lowest in India. In general, the logistic estimation showed that those who are of better socio-economic condition (rich, higher educated, resides in urban areas) are more likely to use

e-cigarette than their counterparts. When adjusted for all other factors, the age-associated estimates of adjusted odds ratio informed no consistent pattern of likelihood across the selected countries.”

Introduction

Consider shortening this section to a page or less and making the language more speculative as it relates to e-cigarettes, as many of the definitively stated comments are either unproven or have been countered in other papers not cited by the authors.

A: Thank you for your suggestion. We have restructured and modified the sentences. Additionally, we have shortened the introduction section with appropriate citations.

Page 4

Line 9: Please cite the source of the stated prevalence estimate “3 million US adolescents”.

A: Thank you for your comment. We have added the appropriate reference to the statement.

Bold, K. W., Kong, G., Camenga, D. R., Simon, P., Cavallo, D. A., Morean, M. E., & Krishnan-Sarin, S. (2018). Trajectories of e-cigarette and conventional cigarette use among youth. *Pediatrics*, *141*(1).

Line 12: consider deleting “the following components”

A: Thank you for your comment. We have deleted “the following components”, and modified the sentence as

“In 2007, the WHO Framework Convention on Tobacco Control (FCTC) proposed a strategy to reduce tobacco use called MPOWER that includes monitoring tobacco use; prevention policies; protecting people from tobacco smoke; providing help to quit tobacco use; warnings about the dangers of tobacco; enforcement of bans on tobacco advertising, promotion, and sponsorship; and raising taxes on tobacco.”

Line 25: Consider rephrasing “e-cigarettes have the potential to be a gateway for quitting smoking”. E-cigarettes are either considered a "gateway for subsequent cigarette smoking" or "a form of tobacco cessation therapy"

A: Thank you for your comment. We have modified the sentence to

“A few studies suggest that e-cigarettes can serve as a gateway for subsequent cigarette smoking.”

Liu X, Lugo A, Davoli E, Gorini G, Pacifici R, Fernández E, et al. Electronic cigarettes in Italy: a tool for harm reduction or a gateway to smoking tobacco? *Tob Control*. 2020 Mar;29(2):148–52.

Bhatnagar A, Payne TJ, Robertson RM. Is There A Role for Electronic Cigarettes in Tobacco Cessation? *J Am Heart Assoc*. 2019 Jun 18;8(12):e012742.

Camenga DR, Kong G, Cavallo DA, Krishnan-Sarin S. Current and Former Smokers’ Use of Electronic Cigarettes for Quitting Smoking: An Exploratory Study of Adolescents and Young Adults. *Nicotine Tob Res Off J Soc Res Nicotine Tob*. 2017 Nov 7;19(12):1531–5.

Hartmann-Boyce J, Begh R, Aveyard P. Electronic cigarettes for smoking cessation. *BMJ*. 2018 Jan 17 [cited 2021 May 26];360:j5543. Available from: <https://www.bmj.com/content/360/bmj.j5543>

Lines 26-33: Can be summarized into one sentence.

A: Thank you for the suggestion. We have rephrased and summarized the sentences.

Line 27: “On the other hand, studies have found that the odds of quitting cigarettes were lower for those who had not formerly smoked” This phrase is confusing, please review. Also, the cited study examined trends in e-cigarette and combustible cigarette use among students in Poland and found that prevalence increased for both. The study was not about quitting smoking, kindly review the reference.

A: Thank you for the comment. The same as the previous comment.

Line 33: the cited study was conducted among adolescents, not young adults. Kindly review the reference.

A: Thank you for the comment. We have corrected the sentence as

“According to a cross-sectional study in USA, e-cigarette use is low among former smokers than the current adult smokers (Mayer et al., 2020).”

Mayer, M., Reyes-Guzman, C., Grana, R., Choi, K., & Freedman, N. D. (2020). Demographic Characteristics, Cigarette Smoking, and e-Cigarette Use Among US Adults. *JAMA network open*, 3(10), e2020694-e2020694.

Line 39: “Although the aerosols from the e-cigarettes are not harmful...” E-cigarette aerosols might contain toxic chemicals and might be harmful. Consider adjusting language.

A: Thank you for the suggestion. We have deleted the statement.

Page 5

Lines 16-37: Consider shortening or eliminating this portion of the introduction.

A: Thank you for the suggestion. We have shortened the suggested lines as

“The diffusion and innovation theory by Everett Rogers, proposed in 1962, suggests that innovations are first appreciated by the upper class, followed by others^{26,27}. E-cigarette use mimics low nicotine, reduces tar exposure, and is more aesthetically appealing than other forms of smoking, making it a more attractive alternative with a higher rate of adaptation²⁸.”

Line 42: “The present study offers a means for the countries involved to inform themselves about young adults’ initialization of e-cigarettes use”. The cited study does not provide information on young adults’ initiation of e-cigarette use, kindly review.

A: Thank you for the suggestion. In the process of shortening the paragraph, we have removed the line.

Data

Page 6

Line 31: Kindly define “PPS” at first mention.

A: Thank you for the comment. We has defined - PPS as “probability proportional to size”

Table 1: Kindly define “HHs” in footnote

A: Thank you for the comment. We has defined - HHs as “Household Surveyed”

Statistical Analyses (Page 7)

Line 25: “further we have used bivariate and multivariate analyses” This sentence appears incomplete. Kindly revise.

A: Thank you for your suggestion. The suggested line has been rewritten in the revised manuscript.

Line 30: Age was not presented in 10-year groups in any of your tables, please revise.

A: Thank you for your comment. The section has been revised entirely based on the comments.

The survey years are different for the various countries e.g. 2011 in Indonesia and 2017 in Ukraine. How do you account for this when making comparisons among countries? This is a critical point that must ne addresses to make sense of this paper.

A: Thank you for your comment. Definitely, the cross-country comparison is not possible as the survey years are not same for all the countries. Instead, using the most recent round of GATS survey for the respective countries, the study demonstrated the most recent pattern of awareness of e-cigarette and its usage in the population which is no way comparative temporally and the study also did not intend to compare the countries.

Since survey data were used in this study, did you apply weights to make estimates nationally representative? If so, please state.

A: Yes, all the country specific estimates are survey weight adjusted. The name of the specific variable is “gatsweight”. Thank you very much for your kind comment.

Also, what statistical test did you use to examine the difference in e-cigarette awareness or use among the different comparison groups?

A: Thank you for your comment. To examine the difference in e-cigarette awareness or use among the different comparison groups, the chi-square test is applied and the corresponding p-value gives the statistical significance.

What p-value determined statistical significance?

A: The statistical significance is determined at level 1%. Thank you very much.

Results

Page 8:

Line 14: kindly include an age group for “younger people”.

A: Thank you for the comment. The dataset contains data for 15 and above years, and we have taken these age groups looking at the sample distribution. If the young age group 15-24 is further divided like '15-19' and '20-24' the sample reduces considerably and not apt for the analysis. Hence, the age variable has been categorized as '15-24', '25-44', '45-64', 'above 64'.

Line 36: “though the association was not significant”...This is confusing. Consider writing out what association you are referring to

A: Thank you for the comment. The sentence has been removed on revision.

Line 41: “...had high usage among the younger age groups”. The prevalence rates reported to support this vary widely. The authors also state in line 46-47 that usage was “high” among older age groups, and report prevalence rates as low as 1.25%. Perhaps state what you consider “high” or consider using a comparator.

A: Thank you for your comment. The comparison was made among the particular age-group across different countries, hence there is variations in values of prevalence rates.

Line 44: from the table, residence was also significant.

A: Thank you for the comment. The sentence has been revised and residence is also added.

Page 9:

Line: 10: Since you used multiple independent variables and one dependent variable (outcome), multivariable is the appropriate term to use.

A: Thank you very much for your kind suggestion. The necessary change has been done in the revised manuscript.

Line 18: Consider replacing “maximum” with “highest”

A: Thank you for the comment. We have removed the table 2 (old manuscript) and hence made corrections in the results section accordingly.

Page 10:

Table 2: Does “NA” imply that variable is missing for that specific country? For instance, there was no data for females in Russia. Does this mean survey was limited to only males? If so, please provide this information under the Data Section.

A: Thank you for your kind suggestion. We have removed the table 2 (old manuscript) and hence made corrections in the results section accordingly.

Page 11:

Table 3: Please define what “***” and “**” mean.

Also, please format the total population and projected users appropriately.

A: Thank you for the suggestion. We have defined *** p<0.01, ** p<0.05, * p<0.10 in the footnotes of Table 2,3,4 and also formatted the total population and projected users on revision.

Discussion

Page 12:

Line 22: Although in Greece, females had higher prevalence of e-cigarette use than males, this was not significant and so I do not think you can state that females were more likely than males to use e-cigarettes.

A: Thank you for the comment. We have deleted the statement from Page 12; Line 22 (old version)

Line 27: Is this a reason why more females use e-cigarettes than males in Greece?

A: Thank you for the comment. We have corrected the statement as

“A study by Tzortzi et al., 2020 has reported that both tobacco smoking and e-cigarette use among females are higher than males, which is attributable to better economic condition.”

Tzortzi, A. (2020). Smoking in Greece. *Pneumon*, 33(2), 59-67.

Line 32: “This raises the seriousness to facilitate initiation of nicotine use at the early stage of life” Not clear, please review

A: Thank you for the suggestion. We have corrected the statement as

“When compared to older people, younger adults in all of the countries studied, use of e-cigarettes are at a higher rate. This emphasizes on enacting new policies to curb the use of nicotine at a young age.”

Line 38: The cited reference [2] does not state this. Kindly review.

A: Thank you for the comment. We have corrected the citation.

Collins, L., Glasser, A. M., Abudayyeh, H., Pearson, J. L., & Villanti, A. C. (2019). E-cigarette marketing and communication: how e-cigarette companies market e-cigarettes and the public engages with e-cigarette information. *Nicotine and Tobacco Research*, 21(1), 14-24.

Line 40: The cited reference [39] does not state this. Kindly review.

A: Thank you for the comment. We have revised the statement as

“In a country like USA, smoke-free laws were enacted before the introduction of ENDS products^{6,49}. Therefore smoke-free laws do not specifically prohibit e-cigarettes and hence people end up using in many places.”

Line 44: “GATS being a cross-sectional survey, we do know whether a former smoker who has successfully given up smoking did so by using e-cigarettes as a cessation method” Please revise.

A: Thank you for the suggestion. We have revised the statement as

“Since the GATS is a cross-sectional study, we can't say whether a former smoker has successfully quit smoking by using e-cigarettes or any other cessation method.”

Line 56: This citation [41] is not needed since you are only stating your results.

A: Thank you for the comment. We have removed the citation [41].

Page 13:

Lines 3-6: This has already been stated on page 12 lines line 24-24.

A: Thank you for the suggestion. We have removed the repetitive statement.

Line 9: “Among countries with higher incomes, such as Costa Rica and Mexico, young adults are more exposed, which may suggest that ecigs users are affected by friends”. How does use in young adults suggest that users are affected by friends?

A: Thank you for the comment. We have corrected the statement as

“Among countries with higher income, such as Costa Rica and Mexico, young adults are more exposed, suggesting a relatively higher availability, affordability and moderate restrictive policies on nicotine vaping products.”

Gravely, S., Driezen, P., Ouimet, J., Quah, A. C., Cummings, K. M., Thompson, M. E., ... & Fong, G. T. (2019). Prevalence of awareness, ever-use and current use of nicotine vaping products (NVPs) among adult current smokers and ex-smokers in 14 countries with differing regulations on sales and marketing of NVPs: cross-sectional findings from the ITC Project. *Addiction*, 114(6), 1060-1073.

Line 37: "Secondly, GATS being the cross-sectional survey we could not able to identify whether the current e-cigarettes user has quit the use before initiation" This is confusing, please revise.

A: Thank you for the comment. We have modified the statement as

"Secondly, GATS being the cross-sectional survey, we do not know whether the current e-cigarettes user had previously stopped conventional smoking."

Conclusion

The authors should consider shortening this section.

A: Thank you for the suggestion. We have modified the conclusion section in more concise form.

"The findings from the present study show that e-cigarette use is widespread, with a higher prevalence among young adults. At the same time, country-specific patterns in the determinants of e-cigarette use, economic wellbeing and higher educational attainment have emerged as consistent predictors of e-cigarette use. With a cross-country analysis of GATS data, this study presents the most recent patterns of e-cigarettes use and its determinants. In addition, it estimates the total number of users based on the recent population so that policymakers and health professionals can be more assertive in their interventions."

Reviewer 2

Dr. Adam Cole, University of Ontario Institute of Technology

Comments for the authors:

This study identifies the prevalence of awareness and use of e-cigarettes among adults in 14 countries. Most surveillance research to-date has focused on developed countries (e.g., US, UK, Canada) and there are few data highlighting the prevalence of e-cigarette use in other countries. The data are relevant for an international audience and can help to inform and evaluate e-cigarette policies. However, I believe there are major edits that are necessary before the manuscript is adequate for publishing. I believe the manuscript would benefit from English language review and editing, and the Methods require additional details. The presentation of the Results could be simplified so they are more easily understood and interpretable.

A: We appreciate the comments, suggestions, and careful reading. We have tried to incorporate the suggestions as best as possible and explain the doubts more clearly. We believe that the current version of the manuscript has improved greatly based on your comments and suggestions.

Major comments

Abstract (page 3, line 11): This statement is incorrect since the analyses did not identify the likelihood of e-cigarette use among different subgroups or characteristics. This would require logistic regression analyses and presenting the results (ORs) from the models.

A: Thank you for the comment. Country specific regression (logistic estimation) has been fitted and the estimated adjusted odds ratio values helped to identify the likelihood of e-cigarette use among different subgroups. The results are presented in the manuscript.

Abstract (page 3, lines 12-16): I'm not sure I understand these points. What cessation behaviour is being referred to – cigarette smoking or e-cigarette use? Since the data are cross-sectional, both behaviours are being measured at the same point in time, so it doesn't seem relevant whether or not the individual quit smoking before using e-cigarettes.

A: E-cigarette use is assumed to be a cessation practice among the current smokers to quit smoking but at the same time we agree that we cannot really infer on individual's quitting of smoking and relate it to the use of e-cigarette given the nature of the dataset being used in this study. Thank you very much for your kind observation and help us to improve the scientific content of the study.

Introduction (page 4, lines 23-53): The second paragraph of the introduction seems to focus on the use of e-cigarettes for smoking cessation. There should be some discussion about the association between e-cigarette use and smoking initiation, particularly among young people given the age range of the sample. Any e-cigarette policy would need to balance the risks of use among young people with the benefits of encouraging smokers to quit.

A: Thank you for the comment. We have added appropriate citation to the modified statement - "A recent study has suggested that young adults who use e-cigarettes had higher odds of conventional smoking initiation. Furthermore, a report published by the National Academies of Sciences, Engineering, and Medicine (2018) had found evidence of an increase in the risk of tobacco smoking due to e-cigarette use among young adults 19. Hence, this poses a major public health challenge that requires strict regulation to access e-cigarettes".

Introduction (page 5, lines 46-51): The authors indicate that IGTC has measured e-cigarette advertising and promotion since 2014. Why haven't the authors included any of these data in their analyses? It would seem these policy data combined with e-cigarette use data from GATS would help to evaluate and inform e-cigarette regulation across countries.

A: Thank you for the comment. The purpose of this study to examine the socio-demographic characteristics for the awareness and use of e-cigarettes across countries, and do not aim to evaluate policies across the countries. Moreover, this is a unit level study to determine the socio-demographic characteristics and we cannot combine these two datasets in this particular study, it would lead to problems in the statistical analysis.

Introduction (page 6, lines 8-18): It seems like a better rationale for the current study is the need for e-cigarette awareness and prevalence data from countries other than the US and the UK (where most current evidence is from). Cross-country data are relevant for evaluating different policy environments.

A: Thank you for your comment. We agree with the reviewer that most of the previous studies are based on USA and UK. This paper finds the gap in the literature look at cross-country awareness and prevalence of e-cigarette use with the best available GATS data which will provide useful clues to policymakers for establishing effective measures for public health planning.

Statistical analysis (page 7, lines 35-39): How were place of residence, wealth index, tobacco users, and advertisement source determined? Do tobacco users include cigarette smokers, smokeless tobacco users, and bidi users? What does “advertisement source” mean? More explanation of how these variables were derived is needed.

A: Thank you for your comment. The section has been revised entirely based on the comments and variable description has been included.

Statistical analysis (page 7): I believe the GATS includes sample weights to account for the sampling design. Did the authors use these when calculating country-specific prevalence rates? This would seem to be a more robust method of calculating country-level prevalence rates.

A: Thank you for the comment. Yes, the sampling weights are used to calculate the country-specific prevalence.

Statistical analysis (page 7, lines 39-40): The authors indicate they calculated country-wide adjusted prevalence rates of current e-cigarette use. How was this done? More explanation is needed to assess whether this is appropriate.

A: Thank you for the comment. We have revise the statistical analysis section on revision

Statistical analysis (page 7): I suggest adding calculations that show the ratio of those who currently used e-cigarettes given they have heard about them [see Zhu et al. (2021). Early adoption of heated tobacco products resembles that of e-cigarettes. *Tobacco Control*. <https://doi.org/10.1136/tobaccocontrol-2020-056089>; or Zhu et al. (2013). The use and perception of electronic cigarettes and snus among the U.S. population. *PloS One*, 8(10), e79332. <https://doi.org/10.1371/journal.pone.0079332>]. These ratios can be compared across countries to show differences in the adoption of e-cigarettes. Despite high awareness, relatively few people in Greece or the Russian Federation report current e-cigarette use, compared with low awareness and low prevalence in India or Ethiopia. The results of this calculation could be added as a new table or described in the text.

A: Thank you for the kind suggestion. Actually, the GATS data contains the question on e-cigarettes based on the fact if they have ever heard about them. Hence, the aware people were only asked about the use of e-cigarettes.

Figure 1: Given that GATS includes sample weights and the survey is nationally representative, 95% confidence intervals should be added to the figure estimates.

A: Thank you for this suggestion, we have added 95% confidence intervals to the Figure 1 on revision.

Supplementary Table 1: I recommend that the supplementary table be included as part of the manuscript, especially since the results of this table are described in the main text and it is the only place where awareness data are presented.

A: Yes, the supplementary table has been included in the main manuscript as Table 2 and 3. Thank you very much for your kind suggestion.

Results (page 9, lines 12-16): The authors indicate that the estimates were adjusted. How was this done? It hasn't been adequately described in the Statistical Analysis.

A: Thank you for your comment. A multivariate analysis always provides the adjusted coefficients of a particular predictor included in the analysis. Thus, it is quite generally and extensively mentioned in the previous literatures. Also, on revision we have added a detailed model description of the statistical analysis on revision.

Table 2: Rather than showing the estimates per 1000 population, it makes more sense to show a prevalence rate (%) since this is easier to interpret and compare. This table also does not show a predicted prevalence, it just shows the prevalence of e-cigarette use at the time of the survey according to sociodemographic characteristics (nothing is being predicted). Changes to this table should be reflected in the results on page 9 (lines 18-34).

A: Thank you for your comment. We have removed predicted prevalence table (old manuscript table 2) on revision.

Table 2: I suggest splitting the youngest age group into 15-18 (youth) and 19-24 (young adults) given concerns about the increasing popularity of e-cigarettes among underage youth.

A: Thank you for your comment. The dataset contains data for 15 and above years, and we have taken these age groups looking at the sample distribution. If the young age group 15-24 is further divided like '15-19' and '20-24' the sample reduces considerably and not apt for the analysis. Hence, the age variable has been categorized as '15-24', '25-44', '45-64', 'above 64'.

Table 2: The age categories do not align with those described in the methods. Why were these age categories chosen?

A: Thank you for your comment. We have corrected and changed the variable description in the methods part on revision.

Table 2: Why aren't education level, wealth index, and advertisement exposure included in this table? Given the discussion of the importance of advertising in the introduction, it seems like this should be included in the table.

A: Thank you for your comment. We have removed table 2 on revision.

Results (page 11, lines 15-23): Again, the authors indicate that the estimates were adjusted, but have not described how this was done.

A: Thank you for your comment. A multivariate analysis always provides the adjusted coefficients of a particular predictor included in the analysis. Thus, it is quite generally and extensively mentioned in the previous literatures. Also, on revision we have added a detailed model description of the statistical analysis on revision.

Table 3: I'm not sure what value this table adds to the manuscript outside of the adjusted odds ratio (but even that is limited). I suggest removing this table.

A: Thank you for your comment. The basic purpose of this table is to show the country-specific total number of current e-cigarette users in the total population.

The **new table 5** has been added incorporating the suggested corrections.

Table 3: I'm not sure why the projected number of total users is included in this table or how it was calculated. It's not actually a projected number of users, but rather an estimated number of users. I suggest removing it as it does not add a lot of value to the manuscript. Any changes should be reflected in the results on page 11 (lines 50-60).

A: Thank you for your comment. Yes, we agree with the comment and it means the estimated number of smokers

Discussion (page 12, line 13): From my review of the manuscript, the analyses did not identify predictors of e-cigarette use across country, so this statement is incorrect and should be removed.

A: Thank you for the comment. We have modified the statement as

"The present study has examined the awareness of e-cigarette use, prevalence, and socio-demographic determinants based on the most recent GATS survey covering 14 countries".

Discussion (page 12, lines 53-54): Again, from my review of the manuscript, the analyses did not identify associations between sociodemographic characteristics and e-cigarette use across countries. The analyses were basically descriptive and did not measure associations. Therefore, this statement is incorrect and should be removed.

A: Thank you for your comment. Actually, we have measured significant association of the socio demographic characteristics with e-cigarette use across countries, by using chi-square test at 1% , 5% and 10% level of significance. Please see revised Table 2 and 3 of bivariate results.

Discussion (page 13, line 35): A major limitation is the range of years included. There have been drastic changes in e-cigarette devices between 2011 and 2017 that need to be acknowledged as they have impacted the brands of e-cigarettes, e-cigarette marketing, and the prevalence of e-cigarette use.

A: Thank you for your comment. We have implemented this suggestion in the revised manuscript.

Conclusion (page 13, line 57): Again, the analyses did not measure associations, so statements about "predictors of e-cigarette use" should be removed.

A: Thank you for the comment. We have rephrased the statement on revision.

Minor comments

Introduction (page 4, lines 25-28): The authors indicate that multiple studies suggest e-cigarettes are a gateway for quitting smoking/prevent people from quitting smoking, but only one reference is provided. I suggest adding in a couple more references (or even a review article) to support your points.

A: Thank you for the suggestion. We have added a few more appropriate references.

Liu, X., Lugo, A., Davoli, E., Gorini, G., Pacifici, R., Fernández, E., & Gallus, S. (2020). Electronic cigarettes in Italy: a tool for harm reduction or a gateway to smoking tobacco?. *Tobacco control*, 29(2), 148-152.

Bhatnagar, A., Payne, T. J., & Robertson, R. M. (2019). Is there a role for electronic cigarettes in tobacco cessation?. *Journal of the American Heart Association*, 8(12), e012742.

Camenga, D. R., Kong, G., Cavallo, D. A., & Krishnan-Sarin, S. (2017). Current and former smokers' use of electronic cigarettes for quitting smoking: an exploratory study of adolescents and young adults. *Nicotine and Tobacco Research*, 19(12), 1531-1535.

Hartmann-Boyce, J., Begh, R., & Aveyard, P. (2018). Electronic cigarettes for smoking cessation. *Bmj*, 360.

Introduction (page 4, lines 39-44): The discussion about marijuana in e-cigarettes is distracting and irrelevant to the current study. I suggest the authors remove it.

A: Thank you for the suggestion. We have deleted the statement.

Introduction (page 4, line 48): I'm not sure what the authors are trying to say here: "The efficacy to quit smoking has relieved the dependency on nicotine."

A: Thank you for the comment. We have removed the statement on revision.

Introduction (page 5, line 21): I'm not sure what the authors mean by "astatically".

A: Thank you for the comment. This was a typo-error and we have corrected the statement.

Data (page 6, line 31): What does "PPS" stand for? This acronym should be spelled out for the reader.

A: Thank you for the comment. We have added the full form of PPS.

Figure 1: The axis should be "Percentage of adults >15 years" (or something similar) since the graph shows both awareness and prevalence of current e-cigarette use

A: Thank you for the suggestion. We have made the change as per the comment.

Results (page 8, lines 27-32): I suggest adding a reference to Figure 1, as it was unclear where these prevalence rates could be found in the manuscript.

A: Thank you for the suggestion. We have added the source.

Source: <https://nccd.cdc.gov/GTSSDataSurveyResources/Ancillary/DataReports.aspx?CAID=2>

Results (page 8, line 56): The authors indicate that current e-cigarette use is higher for current users than for non-users. Users of what product?

A: Thank you for the comment. We have corrected the statement as

“In all countries, the prevalence of current e-cigarette use is observed to be higher for current tobacco users than non-users.”

Table 2: Are the current tobacco smoker results for Mexico correct? It currently shows that e-cigarette use is higher among non-current smokers than current smokers.

A: Thank you for the comment. The bivariate result for the current e-cigarette user in Mexico shows 4.38% among tobacco users. While among non-users it is 0.93% (table 3).

Discussion (page 12, line 44): The line should read “GATS being a cross-sectional survey, we do not know...”

A: Thank you for the suggestion we have corrected the statement and modified the paragraph as well.

“The present study has empirically highlighted the usage of e-cigarettes, which is supposed to be a regulated aid for the cessation of tobacco use. However, this study has some limitations. Firstly, we have considered only 14 countries in the temporal setting of 2011-2017. Secondly, GATS being the cross-sectional survey, we do not know whether the current e-cigarettes user had previously stopped conventional smoking. Lastly, we could not incorporate the drastic changes in e-cigarette devices, brands, marketing between 2011-2017, which might have impacted the prevalence of e-cigarettes use. Nevertheless, despite a few lacunae, there are gaps in the knowledge of cross-country e-cigarette use; this study provides useful clues to policymakers for establishing effective measures for public health planning.”

Reviewer 3

Dr. Allison Glasser, New York University School of Global Public Health

Comments to the Author:

Overall:

This study is one of the first to compare prevalence of awareness of e-cigarettes and e-cigarette use across countries in the WHO regions. Prior to being considered for publication, the manuscript should be edited for English grammar and vocabulary, as there are numerous places throughout the manuscript where the meaning of the sentence(s) was unclear. In addition, the Introduction and Discussion sections need some close editing as per my recommendations below.

A: We appreciate the comments, suggestions, and careful reading. We have tried to incorporate the suggestions as best as possible and explain the doubts more clearly. We believe that the current version of the manuscript has improved greatly based on your comments and suggestions.

Strengths/Limitations:

The 4th bullet ("GATS being a cross-sectional survey, we could not able to identify whether the current e-cigarette user has quit the use before initiation.") is difficult to understand. Consider rewording.

A: Thank you for the comment, we have rephrased and corrected the grammatical error in the sentence. "GATS being the cross-sectional survey, we could not identify whether the current e-cigarette users had previously stopped conventional smoking."

Introduction:

Please provide a citation for the 2nd sentence in the Intro.

A: Thank you for your comment. We have added the appropriate reference to the statement.

Bold, K. W., Kong, G., Camenga, D. R., Simon, P., Cavallo, D. A., Morean, M. E., & Krishnan-Sarin, S. (2018). Trajectories of e-cigarette and conventional cigarette use among youth. *Pediatrics*, 141(1).

The sentence in lines 26-28 (page 4) does not make sense: "On the other hand, studies have found that the odds of quitting cigarettes were lower for those who had not formerly smoked."

A: Thank you for the comment. We have revised the manuscript by shortening the introduction part and in that process had removed these lines.

Lines 28-33, page 4: There are two issues with this statement. First, it is not true – vaping among adult never smokers has never exceeded vaping among current smokers in the US (most recent pub: Margaret Mayer; Carolyn Reyes-Guzman; Rachel Grana; et al Kelvin Choi; Neal D. Freedman. Demographic Characteristics, Cigarette Smoking, and e-Cigarette Use Among US Adults. *JAMA Netw Open*. 2020;3(10):e2020694. doi:10.1001/jamanetworkopen.2020.20694).

Second, the citation at the end of the sentence (citation 9) is a study conducted among adolescents, not adults, and is longitudinal, not cross-sectional.

A: Thank you for the comment. We have corrected the sentence as

"According to a cross-sectional study in USA, e-cigarette use is low among former smokers than the current adult smokers (Mayer et al., 2020)."

Mayer, M., Reyes-Guzman, C., Grana, R., Choi, K., & Freedman, N. D. (2020). Demographic Characteristics, Cigarette Smoking, and e-Cigarette Use Among US Adults. *JAMA network open*, 3(10), e2020694-e2020694.

The sentence in lines 34-37, page 4 ("These chemicals have adverse effects on children, adolescents, and pregnant mothers, which may contribute to cardiovascular disease.") needs a citation.

A: Thank you for the comment. We have added the appropriate citations.

National Academies of Sciences, Engineering, and Medicine. 2018. Public health consequences of e-cigarettes. Washington, DC: The National Academies Press. doi: <https://doi.org/10.17226/24952>.

FCTC, W. (2014). Electronic nicotine delivery systems: report by WHO. 2014. In: October.

Suter, M. A., Mastrobattista, J., Sachs, M., & Aagaard, K. (2015). Is there evidence for potential harm of electronic cigarette use in pregnancy?. Birth Defects Research Part A: Clinical and Molecular Teratology, 103(3), 186-195.

Line 48 (page 4): I don't understand this sentence, "The efficacy to quit smoking has relived the dependency on nicotine."

A: Thank you for the comment. We have revised the manuscript by shortening the introduction part and in that process had removed these lines.

I also do not understand the sentence on lines 21-24 on page 5, "Evidently a study conducted by Pepper et al. [21] found that relatively higher advantageous substitute (e-cigarettes) has higher rate of adaptation."

A: Thank you for the comment. We have modified the statement to make it more meaningful– "E-cigarette use mimics low nicotine, reduces tar exposure, and is more aesthetically appealing than other forms of smoking, making it a more attractive alternative with a higher rate of adaptation".

Starting with line 45 of page 5 ("A handful of studies..."), the remainder of this paragraph is not cohesive nor does it fit with the first half of the paragraph. The statements jump from marketing, to taxes, to cessation, and it is not clear what the point of this section is.

A: Thank you for the comment. We have revised and modified the introduction part.

Results

At the bottom of page 8, do "users" and "non-users" refer to cigarette smokers? If it refers to e-cigarette use, then the sentence does not make sense.

A: Thank you for the comment. Yes, it refers to the cigarette users against non-users. The line has been corrected as "In all countries, the prevalence of current e-cigarette use is observed to be higher among those who smoke cigarette than the non-smokers"

Discussion

I recommend the authors describe the main findings across the bivariate and multivariate models instead of separately, as it feels there is a lot of repetition.

A: Thank you for the comment. This is happening because both the tables provide the prevalence, bivariate is unadjusted and the multivariate is the adjusted. We have revised the analysis part and wrote the results and discussion part accordingly.

There is no discussion of cross-country differences in prevalence in the Discussion section. Could differences be related to policy environments, stage in the tobacco epidemic, etc.?

A: Thank you for the suggestion. We have revised the discussion section.

VERSION 2 – REVIEW

REVIEWER	Michael Blaha Johns Hopkins Hospital, Baltimore, USA, Medicine/Cardiology
REVIEW RETURNED	21-Jun-2021

GENERAL COMMENTS	I think this paper has been markedly improved. Prior to publication, it needs an additional review for grammar and clarity. Additionally, placement of the comma within large numbers need to be review to be consistent with standards in the literature.
--

REVIEWER	Adam Cole University of Ontario Institute of Technology
REVIEW RETURNED	04-Jun-2021

GENERAL COMMENTS	Comments for the authors:
---------------------------

The revised manuscript is an improvement on the initial submission and many of my comments have been addressed by the authors. The revised manuscript would benefit from additional English language review and editing. I have some remaining comments for the authors.

-Title Suggestion: "Awareness and determinants of e-cigarette use across selected WHO region countries: Evidence from the Global Adult Tobacco Survey"

-Abstract (page 1, line 16): remove "patterns" from the Objectives since the manuscript does not describe patterns of e-cigarette use. "Patterns of e-cigarette use" generally refers to different frequencies of use (e.g., daily vs non-daily).

-Abstract (page 1, line 27): add the e-cigarette prevalence for India to be consistent with other results reported.

-Abstract (page 1, line 32-37): the Conclusions should align with the Results reported. Since results for males and young people are not stated in the abstract Results, either remove these from the Conclusions or add a sentence about gender and age differences in e-cigarette awareness and use to the Results.

-Abstract (page 1, lines 33-34): it doesn't appear that your data ask about other ways that people are exposed to e-cigarettes (e.g., through family or friends), so it may not be true that awareness of e-cigarettes came through advertisements in stores or on the internet. Suggest removing this sentence.

-Abstract (page 1, lines 35-36): similar comment to above. Add a sentence to the Results about age differences in e-cigarette awareness and use.

-Introduction: the Introduction is missing an overview of the data about awareness and use of e-cigarettes in different countries. This is necessary to highlight the fact that most prevalence data are from a small selection of high-income countries, and data from other countries are lacking. This naturally leads the reader to the study objectives and the knowledge gaps that this study addresses.

-Introduction (page 3, lines 4-5): this is not the primary source for this prevalence estimate in the United States. The original source should be cited instead. It is also an outdated prevalence estimate.

-Introduction (page 3, lines 23-34): the reference is missing for this sentence.

-Introduction (page 3, lines 28-35): this information seems more relevant to the Discussion than the Introduction

-Methods: the authors have provided a good explanation of the sampling methods of GATS. The authors should include some description of the individual sampling methods (e.g., is the entire household surveyed or is a single individual randomly selected from each household?) and survey methods (e.g., is the survey completed in person/online? Is it an interview or paper/online questionnaire?). What were the response rates for the countries/years that were included in this study? The potential response bias and how it may impact the estimates in the manuscript should be discussed as part of the Strengths & Limitations.

-Methods (page 5, line 8): was a definition or picture of e-cigarettes included with the question? This should be mentioned in the methods. Other data indicate that including definitions/pictures is important for improving the validity of responses among youth.

-Methods (page 5, lines 9-10): add the exact question wording to your description, similar to how awareness of e-cigarettes is shown. Based on the author's response, it also seems like the question about current use was only asked if the participant had heard about e-cigarettes. It was not asked of everyone. This should be noted in the Methods since it would technically affect the denominator for any prevalence estimates.

-Methods (page 5, line 15): replace "smoking e-cigarettes" with "using e-cigarettes" or "current e-cigarette use status". This should be changed throughout the manuscript.

-Methods (page 6, lines 3-9): was the PCA completed by the authors, or is this a variable that is included as part of the GATS dataset? If the authors are describing a procedure that was done by others, they should include an appropriate reference for this.

-Methods (page 6, line 5): check reference for this sentence, as I don't think it is correct

-Methods (page 6, line 9): the authors should include a more explicit description of the categories that were combined for the purpose of their analysis (e.g., "For the current study, we combined the 'poorest' and 'poorer' classes into a 'poor' category, and the 'richer' and 'richest' classes into a 'rich' category").

-Methods (page 6, line 14-19): the questions that were used to assess exposure to advertising seem to more accurately reflect whether or not a participant noticed e-cigarette advertising through these specific channels. I suggest changing the label for these variables throughout the manuscript to more accurately capture the fact that it refers to "noticing" rather than "exposure to" e-cigarette advertising. Participants may have much higher levels of exposure to advertising, but they don't notice it.

-Methods (page 6, lines 21-25): if weighted data were used in the analyses, this needs to be explicitly stated in the manuscript.

-Methods (page 6, lines 30-32): the explanation of what logistic regression is and why it is done is not necessary. Instead, the authors should indicate what statistical package they used to run their models.

-Methods (page 7, lines 3-5): the authors should add more explanation of how they predicted the average e-cigarette use prevalence from the model fit. There are different methods of doing this, so the methods they chose should be explicitly described.

-Figure 1/Table 2/3: add the percentage values to the end of each bar in Figure 1, or add row for the overall awareness and e-cigarette use percentage to the top of Table 2/3.

-Figure 1: the x-axis should be "Percentage of adults >15 years" (or something similar) since the graph shows both awareness and prevalence of current e-cigarette use

-Tables 2, 3, 4: I suggest only showing table results to 2 decimal places unless the journal requirements are otherwise.

-Table 2: a couple chi-square values for Mexico are 0.00. Please verify that these are correct.

-Table 2, 3, 4: generally, $p < 0.05$ is the highest p-value reported rather than $p < 0.10$. I suggest only showing significant associations at $p < 0.05$ or less.

-Table 2 & 3: add a footnote to describe the meaning and use of NA in the table.

-Table 4: if these are adjusted odds ratios that are presented, then 95% confidence intervals should be included, not the standard error.

Not only does this give an indication of whether or not the odds ratio is significant, but it also gives an indication of the level of precision based on how wide the confidence interval is.

- Results (page 13, line 5): should the prevalence be 37/1000 (rather than /100)?
- Table 5: I suggest removing the column of “Current e-cigarette users (estimated)”. It doesn’t add a lot of value since the total populations vary quite widely across countries. It’s not surprising that India has the highest number of users given it has such a large population. I also suggest deleting the associated paragraph from the text (page 13, lines 16-23).
- Discussion: I think an important piece that is missing from the discussion is a comparison of e-cigarette use rates among young adults/adults from the current study to those from other countries not included in this study (e.g., US, UK, Australia, etc.), while taking the year of study into consideration. While there are wide variations in awareness of e-cigarettes (based on data from Figure 1), adult current use rates are all generally low. Are they in line with adult current use rates from other countries and data sets? How do these results support or differ from those of other countries, and what can we learn from these cross-country comparisons?
- Discussion: the authors should be more cautious in their language throughout the Discussion. For example, the prevalence of current e-cigarette use MAY be higher in Russia because of a lack of e-cigarette policies. But the high prevalence of use may also be a result of other factors not measured in this study. Similarly, the lower prevalence of use in India and Vietnam MAY be due to policies in these countries, but it could also be due to other, unmeasured factors. More rigorous studies need to be done to compare cross-country e-cigarette policies and their impact on adult e-cigarette use.
- Discussion: I found the paragraphs to be too long and difficult to follow because they jumped around multiple topics. The flow of the discussion could be improved by focusing each paragraph on an explanation and elaboration of a key result (e.g., a paragraph about differences in awareness between countries, a paragraph about differences in the prevalence of current use between countries, separate paragraphs about similarities/differences in significant sociodemographic predictors (gender, age, education, SES) of current use from multivariate regression models across countries, etc.).
- Discussion (page 14, lines 23-24): It’s not clear how this sentence relates to the discussion that is being made, so I suggest deleting it.
- Discussion (page 14, lines 27-29): I suggest moving the sentence about using e-cigarettes to quit smoking to the discussion about associations between e-cigarette use and smoking, or removing it entirely.
- Strengths & Limitations: an important strength of this study is that it provides important baseline e-cigarette prevalence data for multiple countries. As GATS data collections continue, the prevalence of e-cigarette use in each country can continue to be monitored and used to evaluate policy changes that occur.
- Strengths & Limitations: the authors should comment on potential response bias in the GATS and how this may impact the results of this study (e.g., over- or under-estimate e-cigarette awareness and use)
- Discussion (page 15, lines 20-21): It’s unclear how the reason for

	using e-cigarettes (to quit cigarettes smoking) is relevant to either the fact that the study uses cross-sectional data or to the current study, so I suggest deleting this point. The use of cross-sectional data is an important limitation because it precludes making any kind of causal associations between sociodemographic predictors and e-cigarette use.
--	--

REVIEWER	Allison Glasser New York University School of Global Public Health
REVIEW RETURNED	22-Jun-2021

GENERAL COMMENTS	Overall: The authors have made many improvements to this manuscript based on peer review. I still struggled with some of the spelling/grammar, so prior to being considered for publication, the manuscript should again be edited for English grammar and vocabulary. Throughout: refer to “e-cigarette use” instead of “smoking” e-cigarettes. Introduction: The Introduction may be more impactful if reorganized a bit: e-cigarette use prevalence, evidence on e-cigarettes’ impact on smoking initiation and cessation, what is known about e-cigarette awareness and use in countries globally, and how this study fills gaps. In the first paragraph, I am not sure what relevance the FCTC/MPOWER statement has here. Page 3, lines 23-24: “A recent study has suggested that young adults who use e-cigarettes had higher odds of conventional smoking initiation.” This statement requires a citation. Methods: Outcome variable: Is awareness not also an outcome in this study? Discussion: Page 15, lines 13-15: The statement, “Although the ill-health...” needs a citation.
---

VERSION 2 – AUTHOR RESPONSE

Reviewer #2

Dr. Adam Cole, University of Ontario Institute of Technology

The revised manuscript is an improvement on the initial submission and many of my comments have been addressed by the authors. The revised manuscript would benefit from additional English language review and editing. I have some remaining comments for the authors.

We appreciate to comments, suggestions, and careful reading. We try to incorporate the suggestions as best as possible and explain the doubts more clearly. We believe that the current version of the manuscript has improved greatly based on your comments and suggestions.

Comment 1 : Title Suggestion: “Awareness and determinants of e-cigarette use across selected WHO region countries: Evidence from the Global Adult Tobacco Survey”

Answer: Thank you for the suggestion. We have changed the title of the paper as: *Awareness and determinants of e-cigarette use across selected WHO region countries: Evidence from the Global Adult Tobacco Survey (GATS)*

Comment 2 : Abstract (page 1, line 16): remove “patterns” from the Objectives since the manuscript does not describe patterns of e-cigarette use. “Patterns of e-cigarette use” generally refers to different frequencies of use (e.g., daily vs non-daily).

Answer: Thank you for the comment. We have changed the line as per the suggestion: “The purpose of this study is to describe awareness and prevalence of e-cigarette use by demographic and socio-economic characteristics of the study population (≥ 15 years old) in selected 14 GATS countries.”

Page no: 01; **Line no:** 17-18

Comment 3 :-Abstract (page 1, line 27): add the e-cigarette prevalence for India to be consistent with other results reported.

Answer: As per the suggestion, we have added the e-cigarette prevalence for India (0.7%). Thank you very much.

Page no: 01; **Line no:** 28

Comment 4 :-Abstract (page 1, line 32-37): the Conclusions should align with the Results reported. Since results for males and young people are not stated in the abstract Results, either remove these from the Conclusions or add a sentence about gender and age differences in e-cigarette awareness and use to the Results.

Answer: Yes, we have added gender and age dimension in the result section. Thank you.

Page no: 01; **Line no:** 28-29

Comment 5 :-Abstract (page 1, lines 33-34): it doesn't appear that your data ask about other ways that people are exposed to e-cigarettes (e.g., through family or friends), so it may not be true that awareness of e-cigarettes came through advertisements in stores or on the internet. Suggest removing this sentence.

Answer: Thank you for the suggestion. We have removed the statement.

Comment 6 :-Abstract (page 1, lines 35-36): similar comment to above. Add a sentence to the Results about age differences in e-cigarette awareness and use.

Answer: Thank you for the comment. We have added in the result section

Page no: 01; **Line no:** 29-30

Comment 7 :-Introduction: the Introduction is missing an overview of the data about awareness and use of e-cigarettes in different countries. This is necessary to highlight the fact that most prevalence data are from a small selection of high-income countries, and data from other countries are lacking. This naturally leads the reader to the study objectives and the knowledge gaps that this study addresses.

Answer: Thank you for the suggestion. We have added the gaps of the research.

Page no: 04; **Line no:** 11-19

Comment 8 :-Introduction (page 3, lines 4-5): this is not the primary source for this prevalence estimate in the United States. The original source should be cited instead. It is also an outdated prevalence estimate.

Answer: Thank you for the comment. We have corrected the statement with appropriate recent citation.

Comment 9 :-Introduction (page 3, lines 23-34): the reference is missing for this sentence.

Answer: Thank you for the comment. We have added the appropriate reference to the statement.

Comment 10 :-Introduction (page 3, lines 28-35): this information seems more relevant to the Discussion than the Introduction

Answer: Thank you for the suggestion. We have moved the above statement to the discussion section.

Page no: 15; **Line no:** 17-26

Comment 11 :-Methods: the authors have provided a good explanation of the sampling methods of GATS. The authors should include some description of the individual sampling methods (e.g., is the entire household surveyed or is a single individual randomly selected from each household?) and survey methods (e.g., is the survey completed in person/online? Is it an interview or paper/online questionnaire?). What were the response rates for the countries/years that were included in this study? The potential response bias and how it may impact the estimates in the manuscript should be discussed as part of the Strengths & Limitations.

Answer: Thank you for the comment and suggestion. We have included the description on sampling methods and country specific overall response rates in Table1.

Regarding response bias and the estimation strategy, we have included a para in the discussion section as part of the strengths & limitations section.

Page no: 15; **Line no:** 30 - **Page no:** 16; **Line no:** 4

Comment 12 :-Methods (page 5, line 8): was a definition or picture of e-cigarettes included with the question? This should be mentioned in the methods. Other data indicate that including definitions/pictures is important for improving the validity of responses among youth.

Answer: Thank you for the comment. We have added the definition/description of e-cigarettes as “Electronic cigarettes include any product that uses batteries or other methods to produce a vapor which contains nicotine. They have various other names such as e-cigarette, vape-pen, e-shisha, e-pipes” to the data and method section.

Page no: 05; **Line no:** 05-11

Comment 13 :-Methods (page 5, lines 9-10): add the exact question wording to your description, similar to how awareness of e-cigarettes is shown. Based on the author’s response, it also seems like the question about current use was only asked if the participant had heard about e-cigarettes. It was not asked of everyone. This should be noted in the Methods since it would technically affect the denominator for any prevalence estimates.

Answer: Thank you for the comment. We have modified the statement according to the suggestions. The exact questions to the variables have been mentioned in the methods section. The prevalence estimates were only asked to the respondents who were aware of what electronic cigarettes are.

Page no: 05; **Line no:** 05-11

Comment 14 :-Methods (page 5, line 15): replace “smoking e-cigarettes” with “using e-cigarettes” or “current e-cigarette use status”. This should be changed throughout the manuscript.

Answer: Thank you for the comment. We have made necessary correction throughout the manuscript.

Comment 15 :-Methods (page 6, lines 3-9): was the PCA completed by the authors, or is this a variable that is included as part of the GATS dataset? If the authors are describing a procedure that was done by others, they should include an appropriate reference for this.

Answer: Thank you for the comment. We have elaborated the PCA computation method and have added appropriate citation.

Page no: 06; **Line no:** 03-13

Comment 16 :-Methods (page 6, line 5): check reference for this sentence, as I don't think it is correct

Answer: Thank you for the comment. We have removed the inappropriate reference. *See comment 15*

Comment 17 :-Methods (page 6, line 9): the authors should include a more explicit description of the categories that were combined for the purpose of their analysis (e.g., “For the current study, we combined the ‘poorest’ and ‘poorer’ classes into a ‘poor’ category, and the ‘richer’ and ‘richest’ classes into a ‘rich’ category”).

Answer: Thank you for the suggestion. We have corrected the statement. *See comment 15.*

Comment 18 :-Methods (page 6, line 14-19): the questions that were used to assess exposure to advertising seem to more accurately reflect whether or not a participant noticed e-cigarette advertising through these specific channels. I suggest changing the label for these variables throughout the manuscript to more accurately capture the fact that it refers to “noticing” rather than “exposure to” e-cigarette advertising. Participants may have much higher levels of exposure to advertising, but they don't notice it.

Answer: Thank you for the comment. We have made corrections as per the suggestion.

Comment 19 :-Methods (page 6, lines 21-25): if weighted data were used in the analyses, this needs to be explicitly stated in the manuscript.

Answer: Thank you for the comment. We have added sample weight information in the statistical analysis section.

Page no: 07; **Line no:** 12-13

Comment 20 :-Methods (page 6, lines 30-32): the explanation of what logistic regression is and why it is done is not necessary. Instead, the authors should indicate what statistical package they used to run their models.

Answer: Thank you for the suggestion. We have removed the explanations and added the statistical software used.

Page no: 07; **Line no:** 13

Comment 21 :-Methods (page 7, lines 3-5): the authors should add more explanation of how they predicted the average e-cigarette use prevalence from the model fit. There are different methods of doing this, so the methods they chose should be explicitly described.

Answer: Thank you for the comment. We have provided all the necessary estimation details in the methods section.

Page no: 07; **Line no:** 4-9

Comment 22 :-Figure 1/Table 2/3: add the percentage values to the end of each bar in Figure 1, or add row for the overall awareness and e-cigarette use percentage to the top of Table 2/3.

Answer: Thank you for the comment. We have made the suggested changes in the figure only.

Comment 23 :-Figure 1: the x-axis should be “Percentage of adults >15 years” (or something similar) since the graph shows both awareness and prevalence of current e-cigarette use

Answer: Thank you for the comment. We have made the suggested changes.

Comment 24 :-Tables 2, 3, 4: I suggest only showing table results to 2 decimal places unless the journal requirements are otherwise.

Answer: Thank you for the suggestion. We have made the suggested changes.

Comment 25 :-Table 2: a couple chi-square values for Mexico are 0.00. Please verify that these are correct.

Answer: Thank you for the suggestion. We have corrected the values.

Comment 26 :-Table 2, 3, 4: generally, $p < 0.05$ is the highest p-value reported rather than $p < 0.10$. I suggest only showing significant associations at $p < 0.05$ or less.

Answer: Thank you for the suggestion. We have made the suggested changes.

Comment 27 :-Table 2 & 3: add a footnote to describe the meaning and use of NA in the table.

Answer: Thank you for the suggestion. We have made the suggested changes.

Comment 28 :-Table 4: if these are adjusted odds ratios that are presented, then 95% confidence intervals should be included, not the standard error. Not only does this give an indication of whether or not the odds ratio is significant, but it also gives an indication of the level of precision based on how wide the confidence interval is.

Answer: Thank you for the suggestion. We have made the suggested changes.

Comment 29 :-Results (page 13, line 5): should the prevalence be 37/1000 (rather than /100)?

Answer: Thank you for the comment. We have made corrected the values.

Comment 30 :-Table 5: I suggest removing the column of “Current e-cigarette users (estimated)”. It doesn’t add a lot of value since the total populations vary quite widely across countries. It’s not surprising that India has the highest number of users given it has such a large population. I also suggest deleting the associated paragraph from the text (page 13, lines 16-23).

Answer: Thank you for the suggestion. We have made removed the column and made changes in the manuscript as per the suggestion.

Comment 31 :-Discussion: I think an important piece that is missing from the discussion is a comparison of e-cigarette use rates among young adults/adults from the current study to those from other countries not included in this study (e.g., US, UK, Australia, etc.), while taking the year of study into consideration. While there are wide variations in awareness of e-cigarettes (based on data from Figure 1), adult current use rates are all generally low. Are they in line with adult current use rates from other countries and data sets? How do these results support or differ from those of other countries, and what can we learn from these cross-country comparisons?

Answer: Thank you for the suggestion. We have written an exclusive para on the suggested direction.

Page no: 13; **Line no:** 17 - **Page no:** 14; **Line no:** 07

Comment 32 :-Discussion: the authors should be more cautious in their language throughout the Discussion. For example, the prevalence of current e-cigarette use MAY be higher in Russia because of a lack of e-cigarette policies. But the high prevalence of use may also be a result of other factors not measured in this study. Similarly, the lower prevalence of use in India and Vietnam MAY be due to

policies in these countries, but it could also be due to other, unmeasured factors. More rigorous studies need to be done to compare cross-country e-cigarette policies and their impact on adult e-cigarette use.

Answer: Thank you for the suggestion. We have made changes in the manuscript as per the suggestion.

Comment 33 :-Discussion: I found the paragraphs to be too long and difficult to follow because they jumped around multiple topics. The flow of the discussion could be improved by focusing each paragraph on an explanation and elaboration of a key result (e.g., a paragraph about differences in awareness between countries, a paragraph about differences in the prevalence of current use between countries, separate paragraphs about similarities/differences in significant sociodemographic predictors (gender, age, education, SES) of current use from multivariate regression models across countries, etc.).

Answer: Thank you for the suggestion. We have revised the discussion section with specific paragraphs describing- awareness, use of e-cigarette and differences in significant sociodemographic predictors.

Comment 34 :-Discussion (page 14, lines 23-24): It's not clear how this sentence relates to the discussion that is being made, so I suggest deleting it.

Answer: Thank you for the suggestion. We have deleted the line.

Comment 35 :-Discussion (page 14, lines 27-29): I suggest moving the sentence about using e-cigarettes to quit smoking to the discussion about associations between e-cigarette use and smoking, or removing it entirely.

Answer: Thank you for the comment. We have removed the suggested statement.

Comment 36 :-Strengths& Limitations: an important strength of this study is that it provides important baseline e-cigarette prevalence data for multiple countries. As GATS data collections continue, the prevalence of e-cigarette use in each country can continue to be monitored and used to evaluate policy changes that occur.

Answer: Thank you for the suggestion. We have added the suggested strength to the manuscript.

Page no: 15; **Line no:** 28-29

Comment 37 :-Strengths & Limitations: the authors should comment on potential response bias in the GATS and how this may impact the results of this study (e.g., over- or under-estimate e-cigarette awareness and use)

Answer: Thank you for the comment. We have added the information of potential bias and response rate in the strength and limitation section. See *comment 11*

Comment 38 :-Discussion (page 15, lines 20-21): It's unclear how the reason for using e-cigarettes (to quit cigarettes smoking) is relevant to either the fact that the study uses cross-sectional data or to the current study, so I suggest deleting this point. The use of cross-sectional data is an important limitation because it precludes making any kind of causal associations between sociodemographic predictors and e-cigarette use.

Answer: Thank you for the comment. We have removed the suggested line.

Reviewer # 1:

Dr. Michael Blaha, Johns Hopkins Hospital, Baltimore, USA

We appreciate to comments, suggestions, and careful reading. We try to incorporate the suggestions as best as possible and explain the doubts more clearly. We believe that the current version of the manuscript has improved greatly based on your comments and suggestions.

Comment 1: Prior to publication, it needs an additional review for grammar and clarity. Additionally, placement of the comma within large numbers need to be review to be consistent with standards in the literature.

Answer: Thank you for the comment. Overall, the manuscript has gone through careful revision and language check.

Reviewer # 3:

Dr. Allison Glasser, New York University School of Global Public Health

Overall:

The authors have made many improvements to this manuscript based on peer review. I still struggled with some of the spelling/grammar, so prior to being considered for publication, the manuscript should again be edited for English grammar and vocabulary.

Throughout: refer to “e-cigarette use” instead of “smoking” e-cigarettes.

We appreciate to comments, suggestions, and careful reading. We try to incorporate the suggestions as best as possible and explain the doubts more clearly. We believe that the current version of the manuscript has improved greatly based on your comments and suggestions.

Comment 1: Introduction:

The Introduction may be more impactful if reorganized a bit: e-cigarette use prevalence, evidence on e-cigarettes’ impact on smoking initiation and cessation, what is known about e-cigarette awareness and use in countries globally, and how this study fills gaps.

Answer: Thank you for the suggestion We have revised the introduction and have added statement to fill gaps in the previous studies. See comment # 7 (Reviewer 2)

Comment 2: In the first paragraph, I am not sure what relevance the FCTC/MPOWER statement has here.

Answer: Thank you for the comment. We have deleted the FCTC/MPOWER statement from the introduction section.

Comment 3: Page 3, lines 23-24: “A recent study has suggested that young adults who use e-cigarettes had higher odds of conventional smoking initiation.” This statement requires a citation.

Answer: Thank you for the comment. We have added the appropriate citation.

Comment 4:

Methods:

Outcome variable: Is awareness not also an outcome in this study?

Answer: Thank you for the comment. The main outcome variable is the current use of e-cigarette but we have only checked the prevalence of e-cigarette awareness by socio-economic and demographic characteristics and hence not regressed.

Comment 5: Discussion:

Page 15, lines 13-15: The statement, “Although the ill-health...” needs a citation.

Answer: Thank you for the comment. We have added the appropriate citations.